# Pancreatic cancer intrinsic PI3Kα activity accelerates metastasis and rewires macrophage component

Benoit Thibault[1,2,†] (ID), Fernanda Ramos-Delgado[1,2,†], Elvire Pons-Tostivint[1,2,†], Nicole Therville[1,2], Celia Cintas[1,2], Silvia Arcucci[1,2], Stephanie Cassant-Sourdy[1,2], Gabriela Reyes-Castellanos[3], Marie Tosolini[1] (ID), Amelie V Villard[1,2], Coralie Cayron[1,2], Romain Baer[1,2], Justine Bertrand-Michel[4], Delphine Pagan[1], Dina Ferreira Da Mota[1,5], Hongkai Yan[6,7], Chiara Falcomatà[6,7], Fabrice Muscari[1,5], Barbara Bournet[1,5], Jean-Pierre Delord[1,5], Ezra Aksoy[8], Alice Carrier[3] (ID), Pierre Cordelier[1], Dieter Saur[6,7] (ID), Celine Basset[1,2,5] & Julie Guillermet-Guibert[1,2,*] (ID)

## Abstract

**Pancreatic ductal adenocarcinoma (PDAC) patients frequently suffer from undetected micro-metastatic disease. This clinical situation would greatly benefit from additional investigation. Therefore, we set out to identify key signalling events that drive metastatic evolution from the pancreas. We searched for a gene signature that discriminate localised PDAC from confirmed metastatic PDAC and devised a preclinical protocol using circulating cell-free DNA (cfDNA) as an early biomarker of micro-metastatic disease to validate the identification of key signalling events. An unbiased approach identified, amongst actionable markers of disease progression, the PI3K pathway and a distinctive PI3Kα activation signature as predictive of PDAC aggressiveness and prognosis. Pharmacological or tumour-restricted genetic PI3Kα-selective inhibition prevented macro-metastatic evolution by hindering tumoural cell migratory behaviour independently of genetic alterations. We found that PI3Kα inhibition altered the quantity and the species composition of the produced lipid second messenger $PIP_3$, with a selective decrease of C36:2 PI-3,4,5-$P_3$. Tumoural PI3Kα inactivation prevented the accumulation of pro-tumoural CD206-positive macrophages in the tumour-adjacent tissue. Tumour cell-intrinsic PI3Kα promotes pro-metastatic features that could be pharmacologically targeted to delay macro-metastatic evolution.**

**Keywords** pancreatic cancer; phosphoinositide; PI3K isoforms; targeted therapy; tumour-stroma dialog
**Subject Categories** Cancer; Signal Transduction
See also: GL Raja Arul & ME Fernandez-Zapico (July 2021)

## Introduction

In humans, PI3Ks are composed of 8 isoforms distributed into 3 classes. Each class I PI3K dimers, namely called PI3Kα, PI3Kβ, PI3Kγ and PI3Kδ, are composed of a catalytic subunit (p110α, p110β, p110γ and p110δ) and a regulatory subunit (p85 for α, β and γ, and p101/p87 for γ). PI3Kα and PI3Kβ are ubiquitously expressed. PI3Kγ and PI3Kδ are restricted to the cardiovascular system and leucocytes in normal tissues, but can be overexpressed in solid tumours (Vanhaesebroeck *et al,* 2010). PI3Ks are lipid kinases that phosphorylate phosphatidylinositol 4,5–biphosphate ($PIP_2$) into $PIP_3$ which acts as a second messenger and regulates various functions in normal and tumour cells *via* the PI3K/Akt/mTOR pathway. The PI3K/Akt axis is frequently hyper-activated in cancers and has been tested as a clinical target in recent years (Pons-Tostivint *et al,* 2017; Goncalves *et al,* 2018). PI3K inhibitors are currently described as cytostatic agents, as PI3K activity is

1 Centre de Recherches en Cancérologie de Toulouse, Inserm, CNRS, Université de Toulouse, Toulouse, France
2 LABEX TouCAN, Toulouse, France
3 Aix Marseille Univ, CNRS, INSERM, Institut Paoli-Calmettes, CRCM, Marseille, France
4 Lipidomics, I2MC, Inserm, Toulouse, France
5 Institut Universitaire du Cancer de Toulouse – Oncopole (IUCT-O), Hopitaux de Toulouse, Institut Claudius Regaud Toulouse, France
6 Division of Translational Cancer Research, German Cancer Research Center (DKFZ) and German Cancer Consortium (DKTK), Heidelberg, Germany
7 Chair of Translational Cancer Research and Institute of Experimental Cancer Therapy, Klinikum rechts der Isar, School of Medicine, Technische Universität München, Munich, Germany
8 Centre for Biochemical Pharmacology, William Harvey Research Institute, Queen Mary University of London, London, UK
*Corresponding author. Tel: +33 5 82 74 16 52; E-mail: julie.guillermet@inserm.fr
†These authors contributed equally to this work

critically driving oncogenesis in a cell-autonomous manner. However, the clinical importance of other cell functions regulated by this pathway in a non-cancer cell-autonomous manner, particularly on macrophages, has been underestimated.

Pancreatic ductal adenocarcinoma (PDAC) is a lethal cancer (Neoptolemos *et al,* 2018) where activation of class I PI3K is high and linked to poor prognosis (Schlieman *et al,* 2003). Localised, locally advanced and metastatic PDAC are characterised by early surgical relapse and failure of long-term disease control with chemotherapies. Molecular characterisation of large cohorts of PDAC patients demonstrates that oncogenic *KRAS* mutations on G12 position are found in more than 80% of all patients. There are multiple altered signalling pathways downstream of oncogenic KRAS, including PI3K/Akt pathway (Witkiewicz *et al,* 2015; Conway *et al,* 2019). Fewer than 5% of patients present *PIK3CA* oncogenic mutation, but this mutation mimics the KRAS oncogenic pathway (Eser *et al,* 2013). Our research group and others have demonstrated that the lipid kinase PI3Kα drives the initiation of pancreatic cancer downstream of oncogenic KRAS (Baer *et al,* 2014; Wu *et al,* 2014). However, little is known of the importance of this PI3K isoform in the progression of localised tumours towards a metastatic disease.

Cell-free DNA (cfDNA) and, more precisely, circulating tumour DNA (ctDNA) appears in clinical oncology as an attractive biomarker for early cancer detection, diagnosis and prognosis (Diehl *et al,* 2008; Dawson *et al,* 2013; Alix-Panabières & Pantel, 2016; Abdallah *et al,* 2020). In cancer patients, ctDNA represents a variable fraction of cfDNA (Dawson *et al,* 2013) and is distinguished by the presence of specific cancer-associated mutations. The release of cfDNA can be due to apoptosis and necrosis of cancer cells (or healthy cells), and it can be secreted directly by the tumour or micro-environment cells such as immune and inflammatory cells (Schwarzenbach *et al,* 2011). Because cfDNA has been studied as an exploratory biomarker of micro-metastatic disease in PDAC (Pietrasz *et al,* 2017; Lee *et al,* 2019), we proposed that detection of cfDNA, as a sign of early metastatic disease, could predict therapeutic effectiveness towards metastatic evolution.

Taking two unbiased approaches to analyse patient data, we demonstrated that gene expression signatures of PI3K activation were a novel way to molecularly identify the most aggressive primary tumours. We then identified a novel pharmacological target, PI3Kα, that drives pro-inflammatory features towards macrometastatic evolution. PI3Kα inhibitors could be included in the immunomodulatory and anti-cancer therapeutic arsenal in PDAC.

# Results

## PI3K and PI3Kα-specific transcriptomic signature predicts aggressive pancreatic cancer

We sought to determine in an unbiased manner which signalling pathway are associated with aggressive features in PDAC.

We analysed publicly available data set to distinguish a normal pancreas from chronic pancreatitis (CP) and primary tumours from localised PDAC (PDACloc) or metastatic PDAC (PDACmet) (Fig 1A–C, Dataset EV1). PDACmet and CP patients share the same enrichment of mRNA expression-based hallmarks of biological pathways compared to normal, except for 3 hallmarks. The PI3K/Akt/mTOR

pathway was the most differentially expressed with the lowest *P*-value (Fig 1A). Differential enrichment analysis demonstrates that the PI3K/Akt/mTOR pathway is constantly (equally) enriched in localised and metastatic PDAC compared to normal parenchyma. This trend was confirmed following Reactome pathway analysis; however, we found that PI3K cascade to FGFR2 was significantly increased in PDACmet as opposed to PDACloc, suggesting differential activation of receptor tyrosine kinase (RTK)-coupled PI3K in these samples. Considering that PI3Kα is key for insulin signalling (Vanhaesebroeck *et al,* 2005), angiogenesis (Graupera *et al,* 2008) and PDAC initiation (Baer *et al,* 2014), we then designed a PI3Kα activation gene signature, based on expression levels of PI3Kα-regulated curated genes (Appendix Fig S1A). PI3Kα activation scoring allowed us to cluster 8/9 PDACmet patients (Fig 1C). Conversely, only 2/9 PDACloc clustered with PDACmet. We further validated the PI3Kα activation signature by confirming that cancer cells isolated from peritoneal metastasis (two primary culture of patient-derived cells) presented a significantly higher expression of *FOXA1, RRM2, BIRC5, FGFR4, PHGDH, TYMS,* as well as *MYBL2, PTTG1, KIF2C, CDC20, CCNB1* albeit in a lower extent compared to non-tumoural ductal cells (Appendix Fig S1B). Those cells also presented increased levels of pS473Akt/Akt levels compared to non-tumoural ductal cells (Appendix Fig S1C) analysed by Western blot. This gene subset of the PI3Kα activation signature was increased in all metastatic patients (Fig 1C). We extended our findings to two larger independent cohorts of PDAC patients (Dataset EV2). High scoring of PI3Kα activation was significantly increased in patients with the poorest prognosis, regardless of their stage (Fig 1D). In both cohorts, PI3Kα activation signature was mostly associated with pure basal-like RNA subtype as described in Appendix Fig S2B. The PI3Kα activation signature discriminates between localised patients and those with an early risk of relapse and death (Fig 1E). Even though oncogenic mutations of PI3Kα are rare in PDAC (Eser *et al,* 2013), as confirmed in the PAAD database (Dataset EV3, Appendix Fig S2), high scoring of non-mutated PI3Kα activation was a worse prognostic factor, irrespective of the stage of the disease. In conclusion, activation of non-mutated PI3Kα appears as a strong prognostic factor of aggressive disease.

## Full annihilation of PI3Kα prevents pancreatic cancer cell migration

We then investigated whether pancreatic cancer cells depended on the activity of a specific PI3K isoform. We used a panel of pancreatic tumour cell lines generated by KRAS mutation combined with other genetic alterations including *PIK3CA* mutation and *PTEN* deletion (Dataset EV4A). We compared two different pharmacological strategies of PI3K inhibition using compounds with either isoform-selective or pan-PI3K pharmacological profiles as shown by inhibitory concentration 50 (IC50) on recombinant protein *in vitro* (Fig 2A, Dataset EV4B). We analysed protein expression levels on four cell lines and we observed detectable effects on pS473Akt levels (used as read-out of PI3K activity) for all inhibitors at 1 and 10 μM, thus allowing comparison of the differential downstream actions of PI3K isoforms. The most potent inhibitors, BKM120 and GDC0941, almost annihilated the pS473Akt levels at 10μM in the four cell lines and at 1 μM in only one cell line (R211, data shown in main figure) (Fig 2B–D, Appendix Fig S3) suggesting that effective concentrations (EC50) are

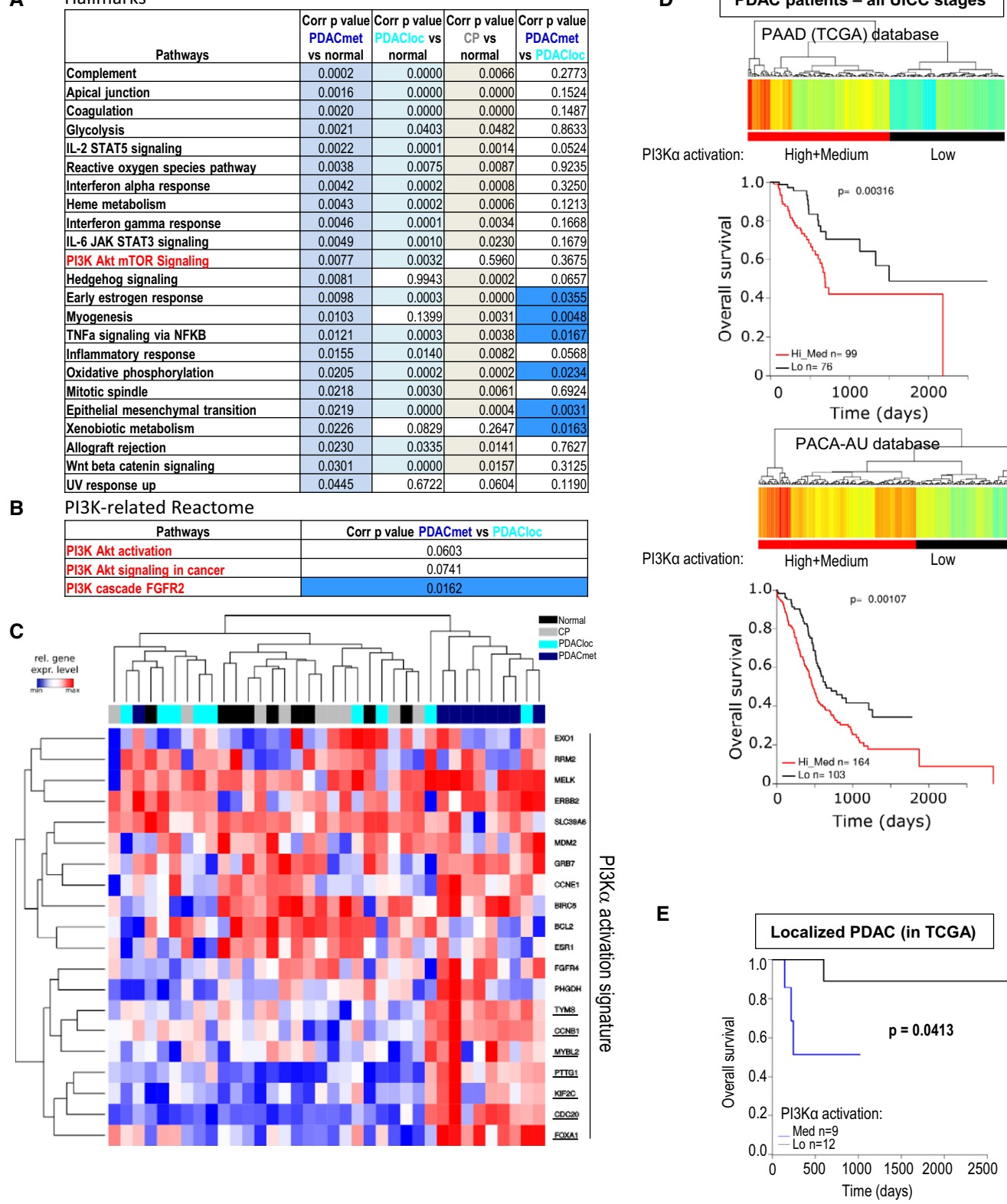

**Figure 1.**

◀

Figure 1. A PI3Kα-specific transcriptomic signature predicts pancreatic cancer aggressiveness.

A   List of 19 HALLMARK pathways (out of 50) significantly altered (ranked with P-value – top: low, bottom: high) in a cohort of 9 metastatic pancreatic adenocarcinoma patients (PDACmet), 9 localised PDAC (PDACloc), 8 chronic pancreatitis (CP) compared to 9 normal pancreas samples (normal); corrected P-values correspond to the comparison indicated. ANOVA test corrected using the Benjamini and Hochberg method (BH) on SES (sample enrichment score). Significantly altered corrected P-values are highlighted in colour. The signature in red corresponds to the signature with the lowest P-value for discriminating PDAC from CP.
B   List of PI3K-related pathway gene signatures (Reactome) altered in the same cohort; corrected P-values are shown. ANOVA test corrected using the Benjamini and Hochberg method (BH) on SES (sample enrichment score). Significantly altered corrected P-values are highlighted in colour.
C   Transcriptomic signature indicative of PI3Kα activation in the same cohort. Genes whose expression is increased only in metastatic patients are underlined. The unsupervised hierarchy of each sample is shown in the top part of the figure; the unsupervised hierarchy of genes is shown on the left; the list of genes is detailed on the right. Blue = low expression, Red = high expression.
D   Scoring of the PI3Kα activation transcriptomic signature was used to cluster patients with high, medium (both groups were combined, high+medium: red) and low (black) scoring levels in the primary tumours of confirmed PDAC patients from PAAD (TCGA) or PACA-AU (BH corrected P-values). The survival curves of each cluster were then plotted, and the statistical significance was calculated using the log rank test.
E   PI3Kα activation scores from localised PDAC in the TCGA database according to their UICC staging were retrieved, clustered into two groups (low and med) (BH corrected P-values) and their overall survival was plotted, with log rank testing.

higher in pancreatic cancer cells compared to *in vitro* IC50. α-selective inhibitors and pan-PI3K inhibitors with low *in vitro* IC50 on PI3Kα significantly decreased pS473Akt levels in the four cell lines at 1 μM, suggesting that inhibitor selectivity profile is conserved.

When α-selective inhibitors, A66 and BYL-719, were used at low concentrations of 0.1 μM, significant effects were observed for assays related to migratory phenotype as shown in Appendix Figs S4 and S5 and in Fig 2E. In all human and murine cell lines tested, A66 and BYL-719 presented a concentration-dependent capacity to inhibit pancreatic cancer cell migratory hallmarks, cell motility (Appendix Fig S4) and directed cell migration (Fig 2E, Appendix Fig S4). The EC30 for migration was reached in 7/9 and 9/9 cell lines for A66 and BYL-719, respectively (Appendix Fig S6B). On the contrary, these parameters were not attained with the γ-specific inhibitor AS252424 or the β and β/δ-specific inhibitors except for AZD8186, which reached the EC30 for migration in two cell lines (Appendix Fig S6B). The R6065 cell line (harbouring *PTEN* deletion) was an exception, in which two β-selective inhibitors had an effect on migration assays at 0.1 μM (Appendix Fig S4B). This demonstrated that motility and migration cell activities are sensitive to PI3Kα inhibition in pancreatic cancer cell. Considering that PI3K cascade to FGFR2 was significantly increased in PDACmet as opposed to PDACloc, FGFR signal activation was prevented in R211 cell line by treatment with 2 μM of AZD4547. AZD4547 significantly decreased R211 cell migration (Fig 2F). The concomitant inhibition of FGFR and PI3Kα by simultaneous BYL-719 and AZD4547 treatment did not inhibit cell migration more than individual treatment which suggested that at least a part of PI3Kα pro-migratory signal was due to FGFR activation. Effects on cell survival and proliferation were significant at higher concentrations (10 μM) (Fig 2G, Appendix Fig S5). We assessed apoptotic R211 cells (with IncuCyte Annexin V assay) after 2 days of treatment with α-specific or pan-PI3K inhibitors and found a selective increase in cell death induced by all PI3Kα inhibitors and by three out of four pan-PI3K inhibitors (Fig 2H). The growth inhibitory GI30 for cytotoxicity was reached in a lower number of cell lines for both A66 and BYL-719 (Fig 2G, Appendix Figs S5 and S6A). Cell migration and cell survival of the non-tumoural ductal pancreatic cell line, HPNE hTERT, were also sensitive to PI3Kα-selective inhibition, but to a lesser extent for the migration assay (Appendix Fig S7).

To validate the pharmacological approach, we used a genetic strategy and treated R211 and PDAC8661 murine pancreatic tumour cells with pools of scramble siRNA, siRNA targeting p110α or mixed siRNA targeting each class I PI3K catalytic subunit, with the latter condition mimicking a pan-PI3K inhibitor. Pools of p110α-targeting siRNAs or pools of p110α/β/γ/δ siRNAs lead to decreased expression of PI3K catalytic subunits (Fig 2I) and induced similar migration inhibition on R211 and PDAC8661 cells compared to pools of scramble siRNAs (Fig 2J). We noticed that, in PDAC8661, decreased expression of p110α mRNA increased p110β and p110δ mRNA expression, without leading to a compensatory increased migration. We also created pools of PANC-1 cells stably transfected with two hairpins targeting *PIK3CA* or two hairpins targeting *PIK3CB* (genes encoding for PI3Kα and PI3Kβ, respectively), as well as one scramble hairpin (Fig 2K). Only shRNA targeting PI3Kα significantly decreased cell migration (Fig 2L). Genetic approaches confirm the selective action of the PI3Kα on the migratory phenotype of pancreatic cancer cells.

### PI3Kα inhibition regulates selective PI-3,4,5-P₃ species

To explain the downstream differences observed with PI3Kα and pan-PI3K inhibitors, we researched a concentration where both inhibitors induced the same effect on Akt phosphorylation (Fig 3A and B). Interestingly, at this concentration, the α-selective inhibitor presented a moderate effect on the number of living cells whereas it drastically affected cell migration; the pan-PI3K inhibitor, BKM120, while being as potent on inhibiting Akt phosphorylation, did not reduce these parameters significantly (Fig 3C and D). This compelling result could be explained by the differential selective decrease of PI-3,4,5-P$_3$ or of PIP$_2$ (comprising both the substrate PI-4,5-P$_2$ and the product PI-3,4-P$_2$) total levels by each inhibitor, respectively (Fig 3E). A66 as opposed to BKM120 selectively reduced the proportion of C36:2 PIP$_3$ (Fig 3F). BKM120 led to the modification of the percentages of other PIP$_3$ species (Fig 3F). The distributions of PIP and PIP$_2$ subspecies were not modified (Appendix Fig S8). Altogether, these data suggest that, in PDAC, PI3Kα-specific actions include production of PIP$_3$ with distinctive acylation pattern, that could explain the selective promotion of cell motility and migration by PI3Kα.

### PI3Kα inhibition targets cell migration and cell survival regardless of the genetic landscape of pancreatic adenocarcinoma cells

PI3Kα oncogenic action is commonly described as directly coupled only to oncogenic KRAS or tyrosine kinase receptors, and not to

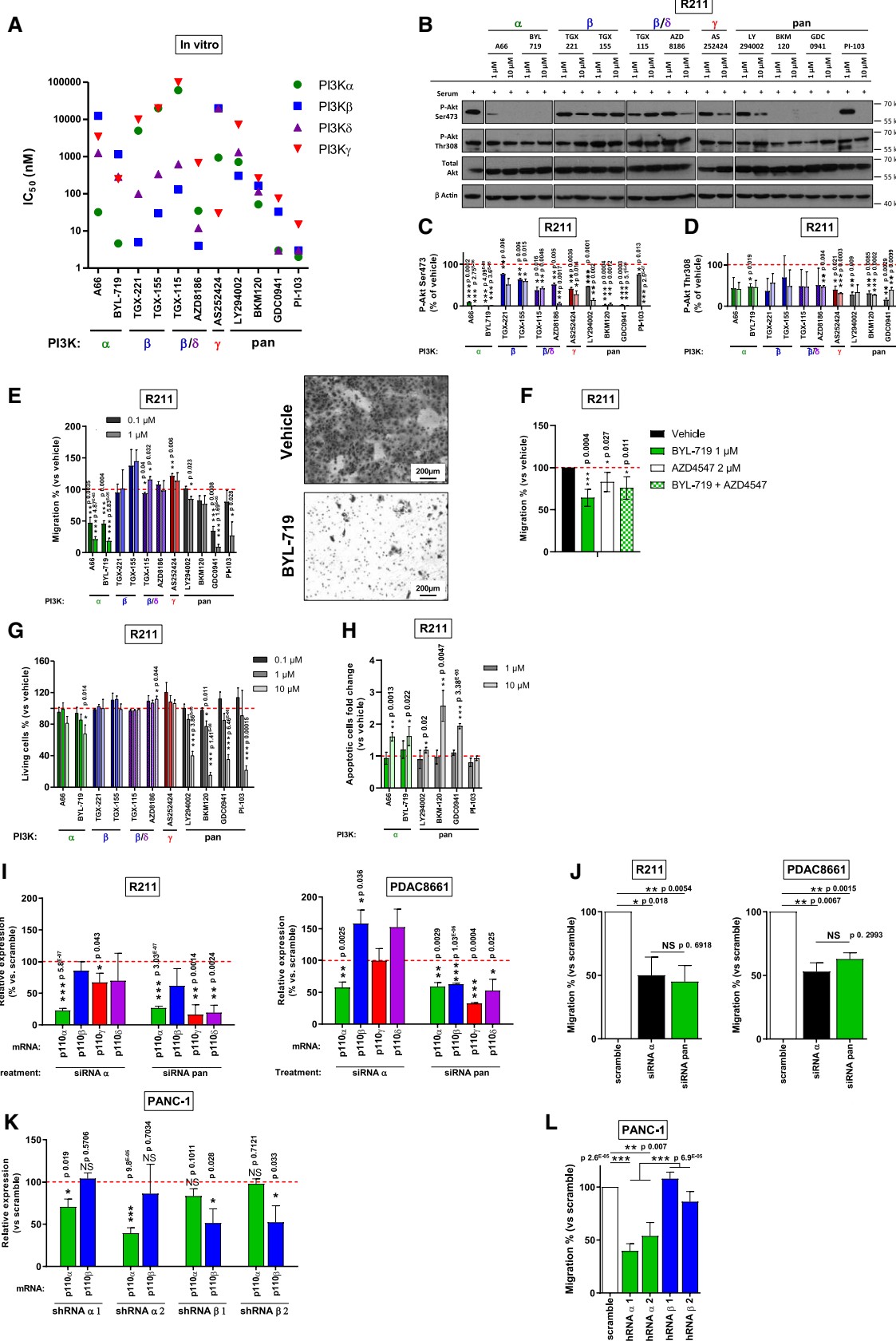

Figure 2.

◀

**Figure 2.   Full annihilation of PI3Kα activity is required to prevent pancreatic cancer cell migration.**

A    *In vitro* IC50 of PI3K inhibitors, obtained on recombinant proteins and based on the literature. The Greek letter on the x axis indicates the most potently targeted isoform.

B–D   (B) Murine pancreatic tumour cells R211 were treated for 15 min with the vehicle, α, β, β/δ, γ-specific or pan-PI3K inhibitors at 1 or 10 μM in the presence of 10% FBS and the protein levels of pAkt (Ser473 and Thr308), total Akt and β Actin were observed by Western blot. PAkt on (C) Ser473 and (D) Thr308 were quantified and normalised with β Actin. n = 2 in each group.

E    Murine pancreatic tumour cells R211 were treated with the vehicle, α, β, β/δ, γ-specific or pan-PI3K inhibitors at 0.1 or 1 μM and simultaneously subjected to a Boyden chamber migration assay. Migrating cells were quantified after 24 h. Representative image of filter after Crystal violet staining is shown. Scale = 200 μm. n = 3 in each group.

F    Murine pancreatic tumour cells R211 were treated with vehicle, BYL-719 1 μM, AZD4547 2 μM or both treatments and subjected to a Boyden chamber migration assay. Migrating cells were quantified after 24 h. n = 4 in each group.

G    Murine pancreatic tumour cells R211 were treated with the vehicle, α, β, β/δ, γ-specific or pan-PI3K inhibitors, and living cells were quantified after 3 days with a MTT colorimetric assay. Metabolically active cells are considered as living cells. n ≥ 3 in each group.

H    Murine pancreatic tumour cells R211 were treated with the vehicle, α or pan-PI3K inhibitors and apoptotic (IncuCyte Annexin V) cells were quantified after 2 days. n = 3 in each group.

I    Relative p110α, β, γ or δ mRNA expression (compared to scramble siRNA) after inhibition of expression by siRNA targeting p110α or a combination of siRNA targeting each class I PI3K isoform. n ≥ 3 in each group.

J    R211 and PDAC8661 cells were treated with siRNA scramble, siRNA targeting p110α or a combination of siRNA targeting each class I PI3K isoform (pools) and subjected to a Boyden chamber migration assay. Migrating cells were quantified after 24 h. n = 3 in each group.

K    Relative p110α or β mRNA expression (compared to scramble shRNA stably transduced cells) in human pancreatic tumour cells PANC-1 stably transduced with shRNA targeting p110α or β. n = 4 in each group.

L    PANC-1-transduced cells were subjected to a Boyden chamber migration assay. Migrating cells were quantified after 24 h. n = 5 in each group.

Data information: Mean ± SEM (*P < 0.05, **P < 0.01, ***P < 0.001, n ≥ 3 independent experiments except for WB experiment, Student's t-test. When not precised, comparisons are performed with vehicle.

*PTEN* alterations (Thorpe *et al*, 2015). To challenge this concept, we analysed the impact of the genetic alterations (on *KRAS*, *PIK3CA*, *PTEN*) and of the organ of origin in determining the role of each class I PI3K in tumour cell migration and cytotoxic sensitivity in response to PI3K inhibitors. With the data shown in Fig 2, Appendix Figs S4E and F, and S5D and E, we calculated the correlation between the *in vitro* IC50 of PI3K inhibitors for all class I PI3Ks (Fig 2A, Dataset EV4B), and their capacity to inhibit migration in nine cell lines (correlation test values are shown in Fig 4A–C, individual values for each cell lines in Appendix Fig S9). The ability of all PI3K inhibitors (at 1 μM) to regulate cell migration mainly depended on their capacity to target PI3Kα, and in a less frequent way on PI3Kβ, δ or γ (Fig 4A, Appendix Fig S9). We reported the *P*-value of the effect versus IC50 correlation test for each class I isoform and found that PI3Kα was significantly associated with pancreatic cell migration (Fig 4B, Appendix Fig S9C), cell motility (Appendix Fig S9A and E), cytotoxicity (Fig 4C, Appendix Fig S9D) and pSer473Akt phosphorylation (Appendix Fig S9B and F). As a positive control, we confirmed the demonstrated isoform dependency that was published by others in other solid and liquid cancer cell lines (Park *et al*, 2008; Torbett *et al*, 2008; Lynch *et al*, 2018). Most cancer cell lines of pancreatic origin depended on PI3Kα activity, and sometimes on PI3Kδ and PI3Kγ to regulate their migration and cell viability despite the genetic context. This result suggests that PI3Kα would be a target of choice for pancreatic cancer patients who, in spite of their genetic heterogeneity associated with class I PI3K activation (mutant *KRAS*, deletion of *PTEN*, mutation of *PIK3CA*), depend on intrinsic basal PI3Kα activity, thus corroborating the prognostic value of the PI3Kα activity signature to detect pro-metastatic features (Fig 1). Epithelial or mesenchymal features of the murine cell line panel were assessed, and cells were classified as either epithelial or mesenchymal phenotype (Appendix Fig S9G). The importance of PI3Kα activity for cell migration and cytotoxic response was similar in both groups (Appendix Fig S9H and I).

## Pharmacological PI3Kα inhibition prevents the rapid progression of cfDNA-positive PDAC

When we compared resected pT3 tumours with and without pathological nodal involvement (pT3N0M0, two tumours versus pT3N1M0, four tumours), we observed a stronger pAkt Substrate IHC staining, indicative of PI3K/Akt activity, in the primary site that are associated with nodal involvement. While PDAC patients with increased levels of ctDNA present a worse prognosis (Pietrasz *et al*, 2017; Lee *et al*, 2019), cfDNA and ctDNA could also be indicative of underlying micro-metastatic disease. Increased pAkt Substrate IHC staining was increased in pT3N1M0 patients with higher level of cfDNA with detected KRAS mutation, which could suggest a correlation with dependency to PI3K activity (Fig 5A and B).

We quantified cfDNA in the KPC model to evaluate the effects of PI3Kα-selective pharmacological inhibition on established tumours presenting high levels of cfDNA. In the KPC model, aggressive pancreatic tumours spontaneously develop under KRAS and p53 oncogenic mutations (Hingorani *et al*, 2005); however, detectable levels of cfDNA have not been described. Tumours were diagnosed through high-resolution US imaging as well as quantification of cfDNA (for the setting of threshold limit, see Materials and Methods below). We quantified the cfDNA in blood plasma samples by longitudinally measuring the relative concentration of two expressed genes, *TP53* and *GAPDH*, and then correlated those findings with the anatomo-pathological results from the pancreas and metastatic site organs (Dataset EV5). The longitudinal average levels of cfDNA correlated with disease progression and mouse lethality (Fig 5C). We analysed two independent cohorts of KPC mice and were able to discriminate between localised primary tumour and metastasis, by cfDNA levels at the time of sacrifice (Fig 5C and D, Appendix Fig S10A–E, Dataset EV5). A small fragment of cfDNA was shown by others to be specific to tumour cells (Thierry *et al*, 2010). Analysis of cfDNA integrity revealed a distinct 160–210 bp fragment selectively increased in mice that developed metastatic PDAC compared

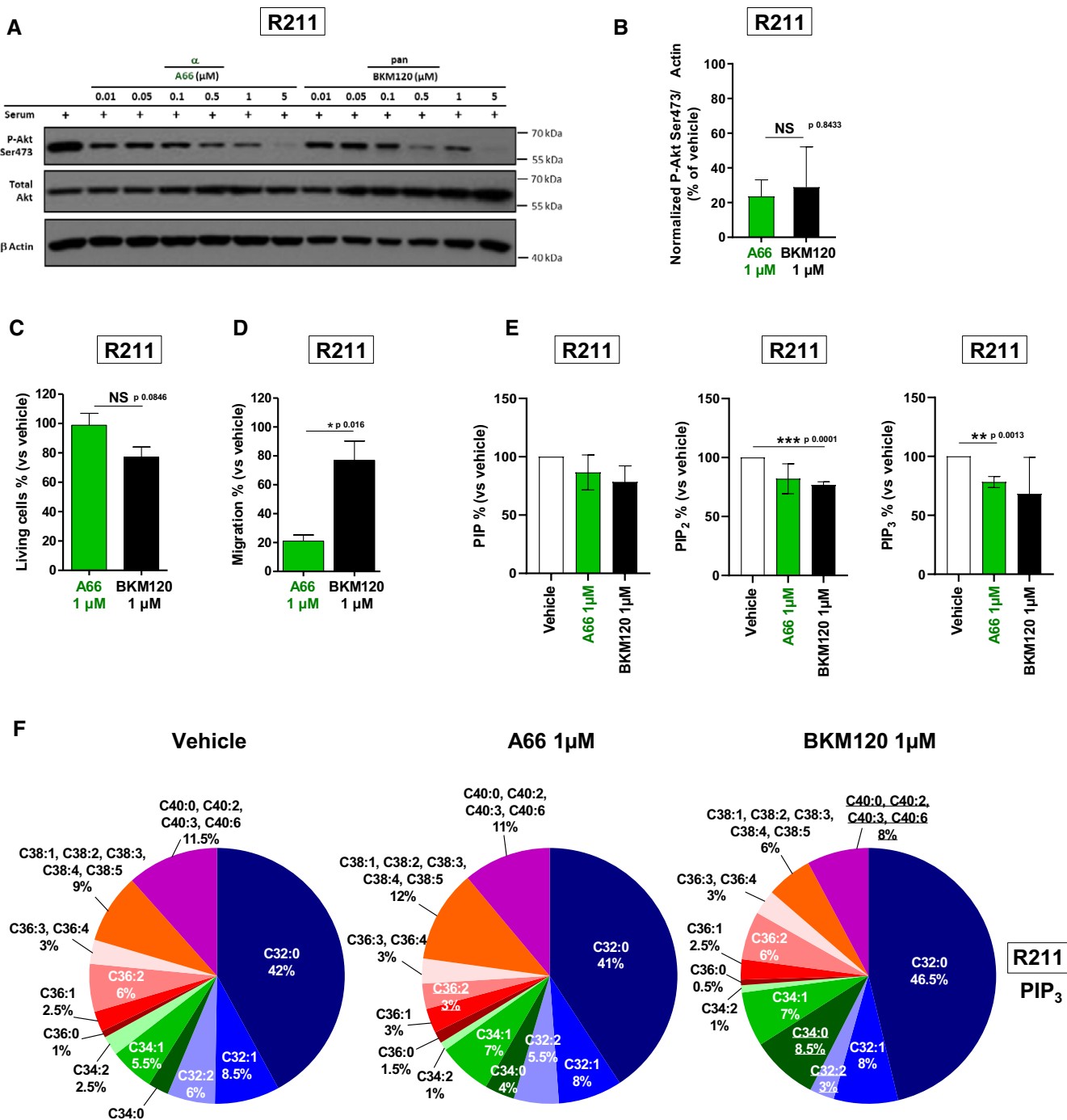

**Figure 3. PI3Kα inhibition regulates selective PI-3,4,5-P3 species.**

A, B  (A) Murine pancreatic tumour cells R211 were treated for 15 min with the vehicle, α-specific A66 or pan-PI3K inhibitor BKM120 at 0.01, 0.05, 0.1, 0.5, 1 or 5 μM in the presence of 10% FBS and protein levels of PAkt (Ser473), total Akt and β Actin were observed by Western blot. (B) PAkt on Ser473 was quantified and normalised with β Actin. $n = 3$ in each group.

C, D  R211 cells were treated with the vehicle, α-specific inhibitor A66 or pan-PI3K inhibitor BKM120 at 1 μM and (C) living cells were quantified after 3 days with a MTT colorimetric assay or (D) migrating cells were quantified after 24 h in a Boyden chamber assay and compared to vehicle (DMSO). $n = 3$ in each group.

E  Murine pancreatic tumour cells R211 were treated for 15 min with the vehicle (0.01% DMSO) or α-specific A66 or pan-PI3K inhibitor BKM120 at 1 μM. Phospholipids were extracted, total PIP, PIP$_2$ and PIP$_3$ were quantified and compared to the vehicle (DMSO). $n = 3$ in each group.

F  The proportion of each PIP3 subtype was represented. Subtypes whose proportion was modified compared to DMSO were underlined. $n = 3$ in each group.

Data information: Mean ± SEM (*$P < 0.05$, **$P < 0.01$, ***$P < 0.001$, $n ≥ 3$ independent experiments, Student's $t$-test).

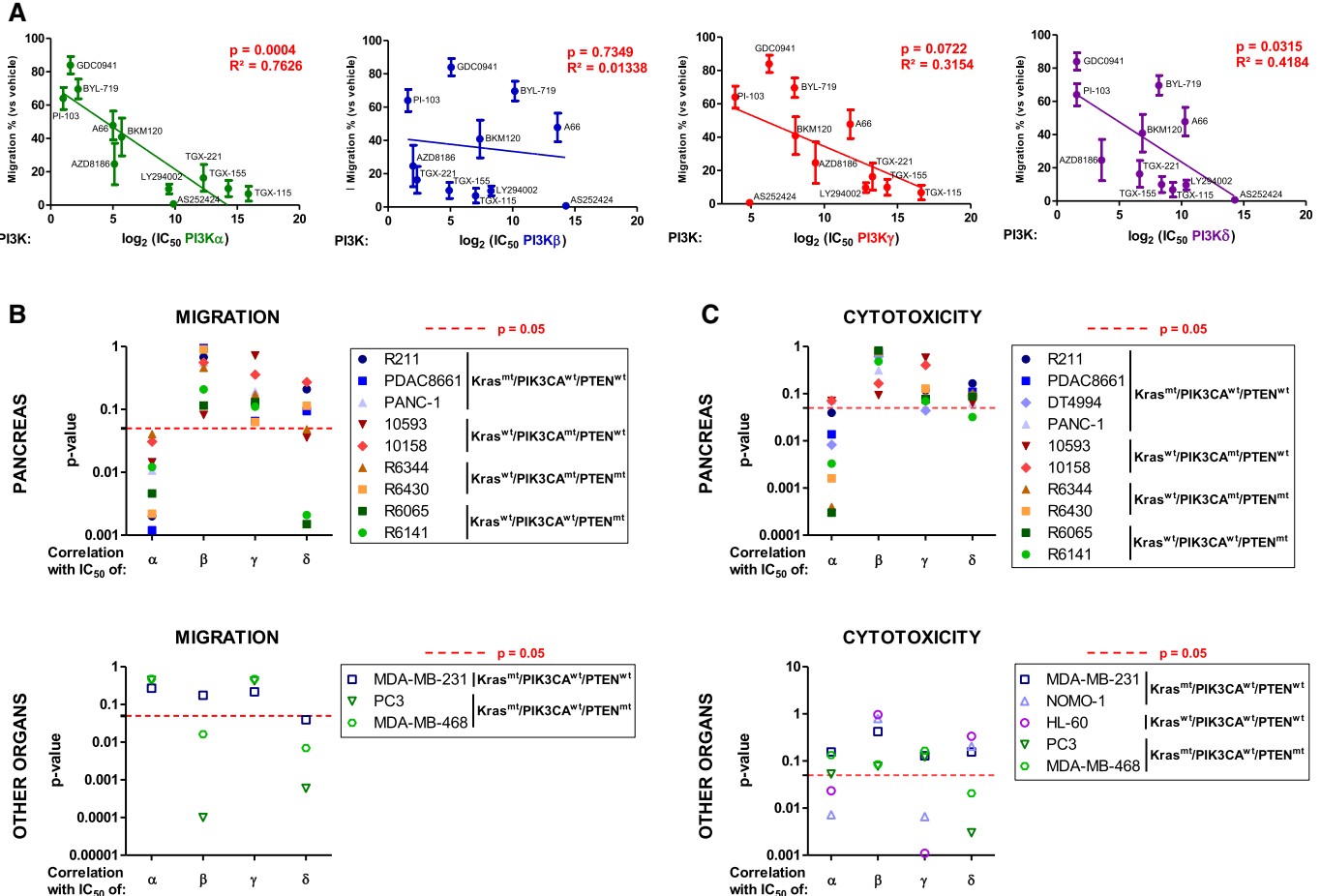

**Figure 4. PI3Kα inhibition is effective on pro-tumoural features regardless of the genetic landscape of pancreatic adenocarcinoma.**

A, B   (A) Migration inhibition capacities of PI3K inhibitors at 1 μM were tested on pancreatic tumour cells (R211, PDAC8661, PANC-1, 10593, 10158, R6344, R6430, R6065, R6141). The mean of these values was plotted against the *in vitro* IC50 for each class I PI3K and each inhibitor. IC50 is determined on recombinant proteins. Pearson correlation tests were performed and its *P*-values were presented separately for each class I PI3K isoform. (B) Individual *P*-values of these correlation tests were represented for each of the 9 pancreatic (R211, PDAC8661, PANC-1, 10593, 10158, R6344, R6430, R6065, R6141,) breast (MDA-MB-231, MDA-MB-468) and prostate (PC3) cancer cell lines; Pearson correlation *P*-values for each PI3K isoform are plotted separately. *n* = 3 in each group.

C   Cytotoxic capacities of PI3K inhibitors at 10 μM were tested on the 10 pancreatic (R211, PDAC8661, DT4994, PANC-1, 10593, 10158, R6344, R6430, R6065, R6141), breast (MDA-MB-231, MDA-MB-468), prostate (PC3) and acute myeloid cells (NOMO-1, HL-60), and the correlation with the *in vitro* IC50 for each class I PI3K of each inhibitor was determined. Pearson's correlation tests were performed, and *P*-values represented for each isoform and cell line. *n* ≥ 3 in each group.

Data information: The dotted red line corresponds to a threshold *P*-value of 0.05 obtained with a Pearson correlation test. Mean ± SEM; *n* ≥ 3 independent experiments.

to mice with high-grade PanINs and localised PDAC (Appendix Fig S10B, Dataset EV5).

We then treated KPC mice featuring aggressive carcinoma (i. e. featuring a US-detected tumour and a high level of cfDNA, Appendix Fig S11), with the PI3Kα-selective inhibitor, BYL-719, or with vehicle (Fig 6A) (André *et al*, 2019). pS473-Akt levels were reduced by BYL-719 treatment in all tested tissues, while pERK-T202 Y204 levels decreased only in the pancreas following BYL-719 treatment (Fig 6B). We interrupted the cohort treatment when vehicle-treated mice presented signs of macroscopic metastatic dissemination, ascites detection, or ethical limit points. The BYL-719 treatment line significantly slowed tumour volume progression (Fig 6C, Appendix Fig S12) and reduced the number of macro-metastatic foci (Fig 6D), distant metastatic area in lungs, liver and

spleen (Fig 6E), delayed ascites development (Fig 6F) and mice survival (Fig 6G). Although most vehicle-treated mice had to be sacrificed during the treatment, only one BYL-719-treated mouse presented signs of tumour evolution that required immediate euthanasia. A correlation was established between the proliferative index of cancer cells (assessed by ki67 index in primary tumour and metastatic sites) and levels of cfDNA, which reflected the global tumour burden (Appendix Fig S13A). BYL-719 significantly reduced cell proliferation assessed by Ki67 index in both primary and meta-static sites (Fig 6H–J, Appendix Fig S13B, for validation of cohort size, see the value of the Power test in Dataset EV5), associated with decreased tumour grading and changes in tumour cell/stroma ratios (Appendix Fig S14). The anticipated secondary effects of BYL-719 treatment on insulin secretion were observed (Appendix Fig S15A

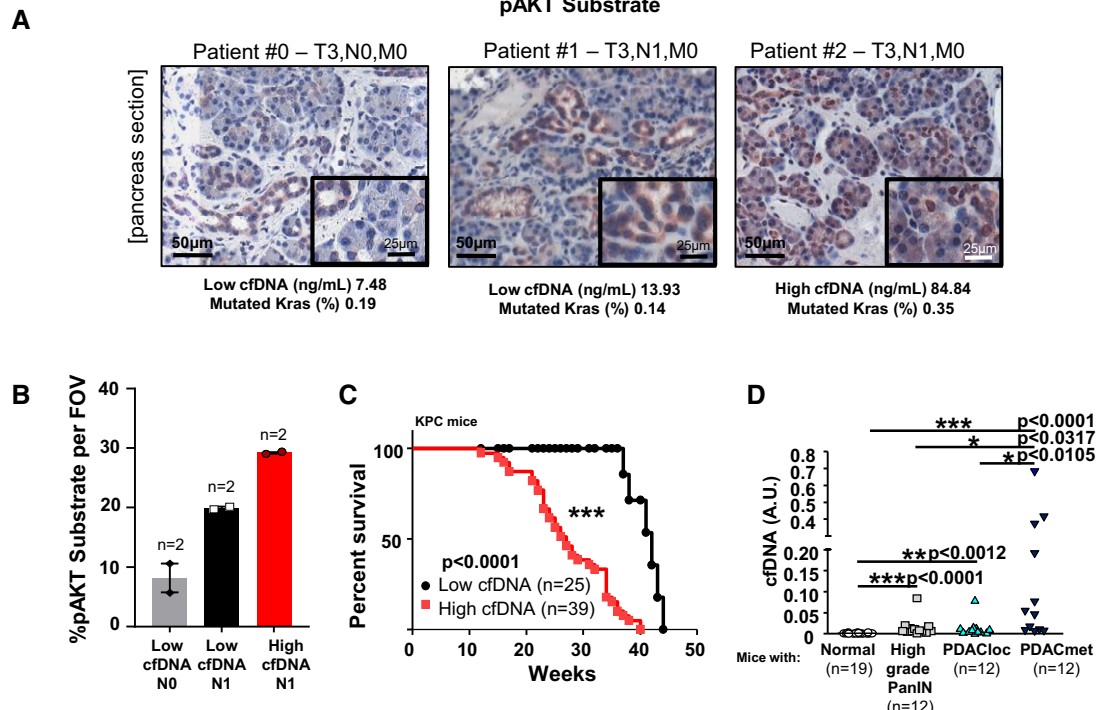

**Figure 5. cfDNA-positive PDAC present increased lethality and metastatic dissemination.**

A Representative images of PI3K activation (as shown by positive pAkt substrate staining) on the primary tumours of PDAC patients, with values of quantification of cfDNA and mutated KRAS allele frequency.

B Quantification of pAkt substrate staining in T3 patients (total number of patients = 6, n = 2 in each group). Mean ± SEM.

C Survival curve of KPC mice presenting different levels of cfDNA. n in each group is indicated. Log rank test, ***P < 0.001.

D Quantification of cfDNA in KPC mice at different stages of the disease. n in each group is indicated. PanIn = Pancreatic Intraepithelial Neoplasia. Non-parametric Mann–Whitney, *P < 0.05, **P < 0.01, ***P < 0.001.

and B). We did not see any significant impact on levels of cleaved caspase 3-positive cells (apoptotic marker: Appendix Fig S15C) or on γ-H2AX-positive foci (DNA damage marker: Appendix Fig S15D).

Furthermore, in a context of early pancreatic cancer lesions induced by oncogenic KRAS in an inflammatory condition (caerulein injections), the highly selective pharmacological inactivation of PI3Kα with another compound, namely GDC-0326, completely prevented the maintenance of pre-cancer lesions (Appendix Fig S16A–D), and features linked to stromal remodelling (Appendix Fig S16B and E). The high level of cleaved caspase 3 detected in epithelial lesions could be prevented by PI3Kα inactivation (Appendix Fig S16B and F).

Tail vein injection of murine pancreatic cancer cells, engineered to express secreted luciferase (R211-Luc cells) in Nude mice, was carried out in order to confirm the action of BYL-719 treatment on the evolution of micro-metastatic foci. Administration of BYL-719 treatment for 21 days significantly prevented tumour cell growth in lungs (Fig 6K and L, Dataset EV5). Interestingly, vehicle-treated mice presented an increased area of pancreatic epithelial marker CK19 (Fig 6M). Similar results were observed in C57/B6 mice when comparing the area of metastatic foci after tail vein injection of syngeneic Kras mutant pancreatic cells (KC;p110α$^{+/+}$, A338) compared to syngeneic Kras mutant pancreatic cells partly lacking PI3Kα activity through a genetic inactivation of one allele of *Pik3ca*

(KC;p110α$^{+/lox}$, A260, see also Appendix Fig S17, Appendix Supplementary Methods) (Fig 6N). Taken together, these data demonstrate that *in vivo* inhibition of PI3Kα prevents the evolution of micrometastatic foci into macro-metastatic foci and the rapid progression of cfDNA-positive PDAC.

### Tumour-intrinsic PI3Kα alters tumour cell chemokine secretion and promotes the acquisition of tumour-associated inflammatory macrophage characteristics in the peritumoural tissue

Transition from micro- to macro-metastasis is now well described as promoted by tumour-extrinsic factors including immune cells (Celià-Terrassa & Kang, 2016). Hence, we first analysed systemic alterations on circulating blood cells during PDAC progression in our KPC model. We performed a full blood count to assess immune cell populations (Dataset EV5), as macrophages promote PDAC progression (Padoan *et al*, 2019). We noticed that metastatic KPC mice presented significantly increased counts of white blood cells (Fig 7A), monocyte (Fig 7B) and granulocyte (Fig 7C) counts compared to KPC with localised tumours. The lymphocyte counts remained the same (Fig 7D).

To test whether tumour-intrinsic PI3Kα could be involved in this tumour/stroma interaction, we used derived cell lines from

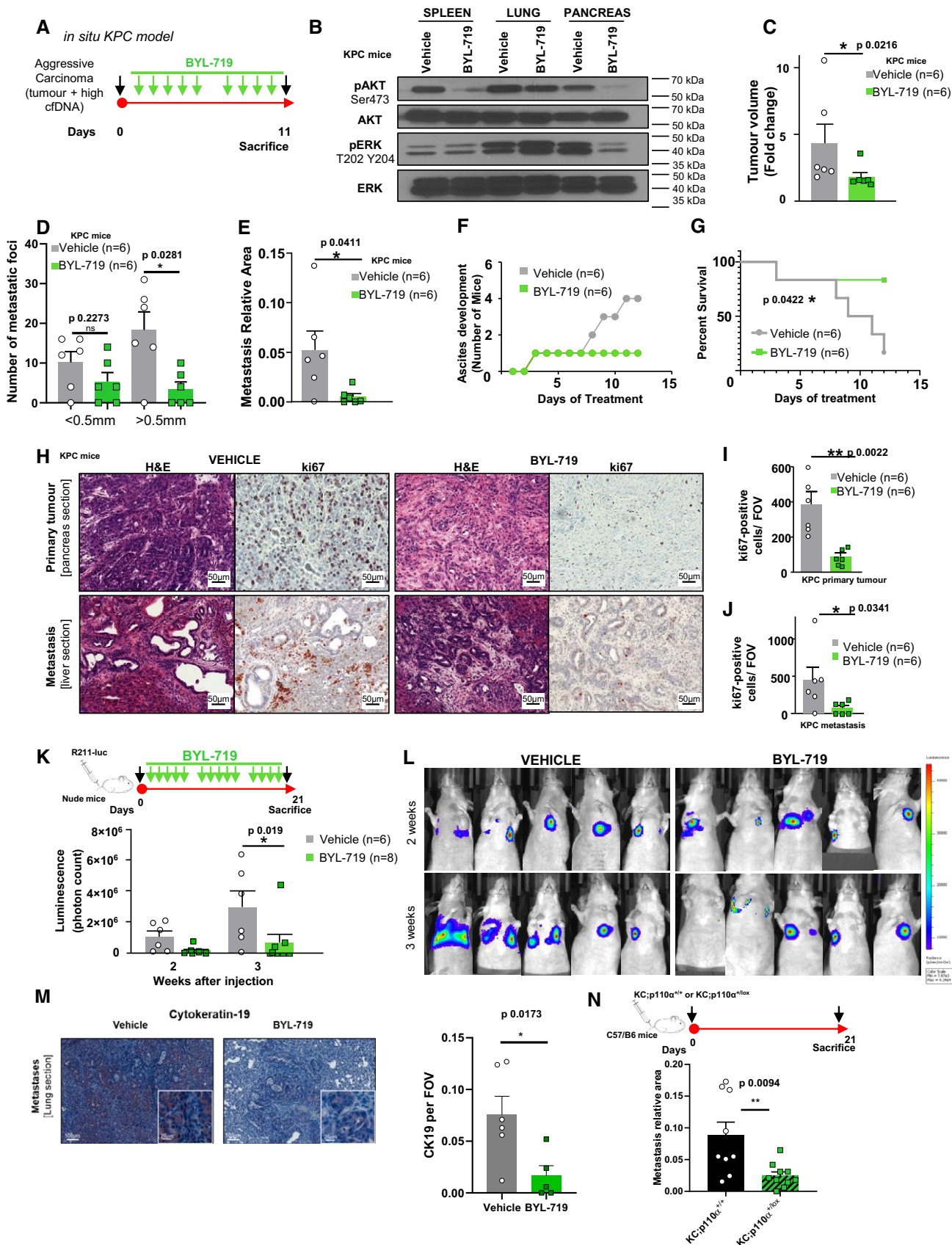

**Figure 6.**

**Figure 6. Pharmacological PI3Kα inhibition prevents the rapid progression of cfDNA-positive PDAC.**

A   KPC mice diagnosed with an aggressive carcinoma were given daily oral doses of BYL-719 (50 mg/kg) or of vehicle (0.5% methyl cellulose with 0.2% Tween-80). $n = 6$ in each group

B   The protein levels of pAkt (Ser473), total Akt, pERK (Thr202 and Tyr204) and total ERK were observed by Western blot in spleen, lung and pancreas tissue lysates after treatment with oral doses of vehicle (0.5% methyl cellulose with 0.2% Tween-80) or of BYL-719 (50 mg/kg).

C–E  (C) Quantification of tumour volume (expressed in fold change) and (D, E) quantification of micro- and macro-metastatic foci and the relative metastasis area (comprising metastases in the spleen, liver and lung) of KPC mice treated with oral doses of vehicle (0.5% methyl cellulose with 0.2% Tween-80) or of BYL-719 (50 mg/kg).

F   Development of ascites in KPC mice after treatment with oral doses of vehicle (0.5% methyl cellulose with 0.2% Tween-80) or of BYL-719 (50 mg/kg).

G   Survival curve of KPC mice treated with the vehicle or BYL-719. Log rank test, *$P < 0.05$.

H–J  (H) Representative images and (I) quantification of Ki67-positive cells in pancreatic tumours and in (J) metastasised liver sections after treatment with oral doses of vehicle (0.5% methyl cellulose with 0.2% Tween-80) or of BYL-719 (50 mg/kg).

K, L  (K) Treatment regimen of the tail vein injection experiment in nude mice treated with the PI3Kα inhibitor or vehicle ($n$ is indicated), quantification at two time points and (L) representative images of luminescence via IVIS® Spectrum *in vivo* imaging system.

M   Quantification and representative CK19 IHC of R211-luc lung micro-metastasis from vehicle or BYL-719-treated mice. $n = 6$ in vehicle; $n = 5$ in BYL-719 group.

N   Tail vein injection experiment in C57/B6 mice of murine syngeneic pancreatic cancer cells (Kras mutated A338, Kras mutated and half PI3Kα inactive, A260; $n > 8$ in each group), quantification of micro- and macro-metastasis at final time point.

Data information: Mean ± SEM (*$P < 0.05$, **$P < 0.005$) C–E, I–J, M, N: non-parametric Mann–Whitney; G: log rank test.

tumoural lesions induced either by Kras mutant or Kras mutant in pancreatic cells partly lacking PI3Kα activity, presenting differential metastatic potential (Fig 6N). We found that the genetic inactivation of PI3Kα in pancreatic cancer cells led to an altered cytokine secretion pattern *in vitro* (a panel of 16 cytokines were tested), with decreased levels of IL-3 in two mutant Kras pancreatic cancer cell lines partly lacking PI3Kα activity (genetic inactivation) compared to three mutant Kras pancreatic cancer cell lines, including R211 cells (Fig 7E, Appendix Fig S17). Pharmacological inactivation of PI3Kα in three different mutant Kras pancreatic cancer cell lines significantly and reproducibly decreased IL3 levels (Fig 7F). PDAC patients with a high PI3Kα activity signature presented a significant increase on the gene signature for the selective immune population of γδ T lymphocytes (LTγδ) and not on generic signatures of other immune cell populations (Fig 7G, Dataset EV2C). The γδ T lymphocyte signature is known to be associated with poor prognosis and highly inflammatory conditions, promoting differential macrophage differentiation (Daley *et al*, 2016).

We therefore analysed immune cell composition in the cohort of KPC mice treated with BYL-719 or vehicle. BYL-719 did not modify the overall F4/80$^+$ macrophage count (Fig 7H and I), but significantly prevented their differentiation into pro-tumourigenic CD206$^+$ macrophages (tumour-associated inflammatory macrophages) in tumour-adjacent tissues (Fig 6H and J), with no significant difference in CD4$^+$ and CD8$^+$ infiltration (Appendix Fig S18). To test whether oncogenic PI3Kα also mimicked oncogenic Kras on tumour/ stroma interaction, we assessed F4/80$^+$ macrophage recruitment and infiltrating CD206$^+$-macrophages around tumours and found them at a similar rate in peritumoural tissue from hyper-activated oncogenic PI3Kα (p110α$^{H1047R}$) and KRAS mutant tumours (Fig 7K and L). Oncogenic PI3Kα did not trigger inflammation in normal pancreas of young mice (1–3 months old) as assessed by quantifying infiltrating F4/80$^+$ macrophages (Appendix Fig S18). Inactivation of PI3Kα exclusively in pancreatic epithelial cells was sufficient to completely prevent CD206$^+$ macrophage infiltration around high-grade lesions (Fig 7M and N, Appendix Fig S18C and D). Increased peritumoural CD206$^+$ staining is associated with further development of metastatic foci and ascites (Fig 7O, Appendix Fig S18E).

To assess macrophage function when tumoral PI3Kα is inhibited, conditioned medium from pancreatic tumour cells treated or not

with BYL-719 or with genetic PI3Kα inactivation were used to assess cytokine production of IC21 macrophage cell line (Fig 7P). From the tested cytokines, only TNFα secretion was significantly decreased by both pharmacological and genetic inactivation of PI3Kα (Fig 7Q and R). IL-3 blocking antibody was added to the conditioned medium and decreased TNFα production by IC21 macrophage cells compared to control antibody-treated cells (Fig 7R). Exogenous TNFα (20ng/mL) promoted R211 pancreatic cancer cell migration and PI3Kα inhibition by BYL-719 inhibited cell migration induced by TNF-α (Fig 7S).

Our data show that the rapid progression of aggressive cfDNA-positive PDAC driven by PI3Kα involves changes in the inflammatory cytokine context in conjunction with inflammatory pro-tumoural macrophage characteristics in peritumoural tissue. Tumour-induced increased TNFα secretion by macrophages could further promote PI3Kα-driven tumour cell migration.

## Discussion

In the path of the recent success of targeted therapies in pancreatic metastatic patients (Cintas *et al*, 2018; Golan *et al*, 2019), further attempts should be made to prevent their rapid progression. We have demonstrated that such strategy could be to target the PI3Kα-driven signal that critically sustains several aspects of pancreatic oncogenicity, including those at the origin of tumour-induced environment rewiring and metastasis evolution.

In the clinical setting, single agents targeting PI3K had a limited impact (Shapiro *et al*, 2014), mainly due to the lack of selection of patients with advanced disease (Le Tourneau *et al*, 2015). In metastatic breast cancer (MBC) patients, when *PIK3CA* mutation was detected in circulating tumour DNA, progression-free survival (PFS) improvement became largely significant (Delaloge & DeForceville, 2017), and BYL-719 (alpelisib) was recently granted marketing authorisation for the treatment of breast cancer in combination with endocrine therapy (André *et al*, 2019). Our study suggests that contrary to MBC, even without *PIK3CA* oncogenic mutation, the PI3K pathway could be a driver of pancreatic metastatic evolution, therefore, a druggable target. Patients with PI3Kα activation signature are enriched in pure basal-like phenotype. This RNA-based

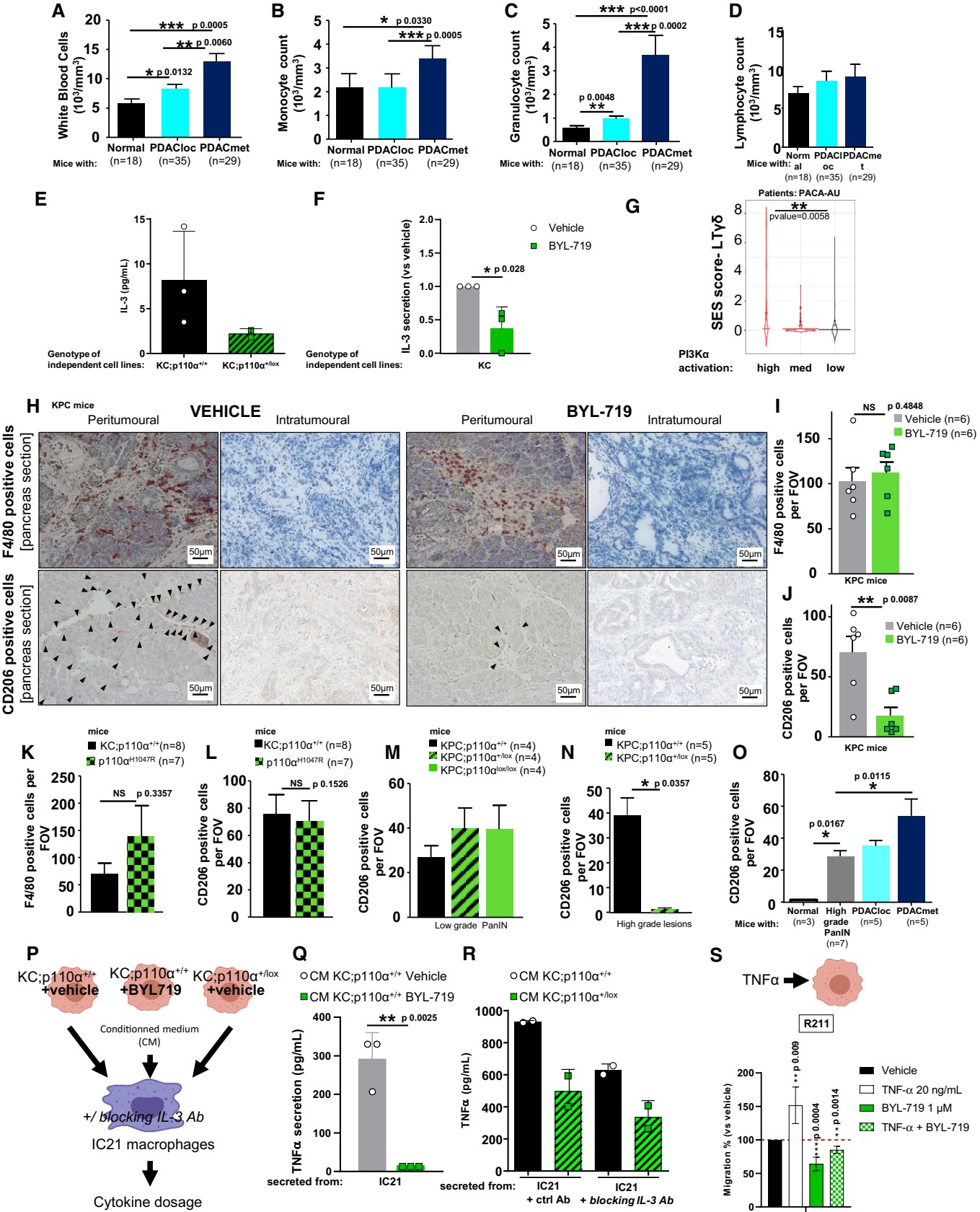

Figure 7.

**Figure 7.  Tumour-intrinsic PI3Kα alters tumour cell chemokine secretion and promotes the acquisition of tumour-associated inflammatory macrophage characteristics in the peritumoural tissue.**

A–D     Blood count of (A) white blood cells, (B) monocytes, (C) granulocytes and (D) lymphocytes in KPC mice, with cohort size in each group.
E       Basal level of IL-3 was quantified in KC;p110α$^{+/+}$ (R211, A338 and A338L) and KC;p110α$^{+/lox}$ (A260 and A94L) cell lines isolated from mice primary tumours or lung metastases (L). Each point is a mean of $n$ = 3 independent values.
F       IL-3 relative level (compared to vehicle) was determined in 3 KC (R211, A338 and A338L) cell lines isolated from mice primary tumours under treatment with vehicle (DMSO) or 1µM of BYL-719. $n$ = 3 independent values.
G       Violin plot demonstrating the link between PI3Kα activation and activation of a selective population of γδLT in the PACA-AU cohort.
H–J     (H) Representative images and quantification of (I) F4/80-positive cells and (J) CD206-positive cells after the pharmacological inhibition of PI3Kα. Black arrowheads show positive immune cells.
K, L    Quantification of (K) F4/80 and (L) CD206-positive cells in KC and oncogenic PI3Kα mice.
M, N    Quantification of CD206-positive cells in (M) low-grade and (N) high-grade PanIN lesions after the genetic inactivation of PI3Kα in epithelial pancreatic cells.
O       Quantification of CD206 positive in KPC mice at different stages of PDAC progression.
P       Treatment of macrophage cell line with tumour cell-conditioned medium.
Q, R    Quantification of secreted TNFα in indicated conditions. $n$ = 3 independent values for each cell line and treatment.
S       Murine pancreatic tumour cells R211 were treated with vehicle, TNF-α 20 ng/ml, BYL-719 1 µM or both treatments and subjected to a Boyden chamber migration assay. Migrating cells were quantified after 24 h. $n$ = 4 independent values.

Data information: n in each group is indicated in the figure. A-D, I-O, Mann–Whitney; F, Q-S, Student $t$-test; G, ANOVA test corrected using the Benjamini and Hochberg method (BH) on SES (sample enrichment score). Mean ± SEM (\*$P$ < 0.05, \*\*$P$ < 0.005, \*\*\*$P$ < 0.0001).

PDAC subtype is also known as the most aggressive subtype of PDAC patients (Collisson *et al*, 2011; Puleo *et al*, 2018). Mueller *et al* (2018) showed that this subtype of murine cell lines showed a strong gene set enrichment for Ras downstream signalling pathways, including PI3K/Akt signalling, further corroborating our finding.

In PDAC, high levels of ctDNA as a marker of micro-metastatic disease were correlated with poor prognosis (Pietrasz *et al*, 2017). Experimental data from other investigators also argue that disseminating cells are detected very early in the disease history (Rhim *et al*, 2012). Our longitudinal analysis of cfDNA levels demonstrates an early increase of these parameters in an experimental pancreatic cancer model. Quantification of the concentration of the 160–210 bp fragment increases the prediction rate of survival, as evidenced in patients in other studies using circulating mutation rates (Mouliere *et al*, 2018).

In terms of toxicity, the main concern for using potent PI3Kα inhibitors remains the induction of insulin feedback which could feed the tumours (Goncalves *et al*, 2018). These concerns could be resolved by the clinical management of glycaemia during treatment (Rodon *et al*, 2013; Goncalves *et al*, 2018). It has to be noted that this insulin feedback occurs to be reduced by age and that PDAC is mostly detected in patients > 40 years (Foukas *et al*, 2013).

Our data also demonstrate that oncogenic KRAS-PI3Kα coupling leads to specific functions of clinical relevance in the pancreatic context. In mutant KRAS-driven lung cancers, inactivation of PI3Kα yielded only a partial response (Castellano *et al*, 2013), MAPK pathway activity being key in this context. From our results, there are four non-exclusive explanations of the importance of KRAS-PI3Kα coupling in this organ setting.

Firstly, in the pancreas, PI3Kα selective inactivation decreased the phosphorylation of Erk1/2. Our data are not in line with data from other studies in this regard (Alagesan *et al*, 2015; Junttila *et al*, 2015); other authors used pan-PI3K inhibitors. MEK inhibition has a minimal anti-tumoural action due to the induction of strong feedback activation on Akt (Kong *et al*, 2016). Selectivity towards PI3Kα could prevent the induction of compensatory signals towards the MAPK pathway.

Secondly, the action of PI3Kα on actin cytoskeleton remodelling in pancreatic cancer cells is sensitive to pharmacological inhibition.

PI3Kα inhibitor is efficient regardless of the mesenchymal/epithelial status of the tested cell lines. In PDAC, an epithelial-to-mesenchymal transition-independent metastasis programme is identified that promotes macro-metastatic foci (Chen *et al*, 2018). Macro-metastatic foci are enriched in E-cadherin expressing tumour cells in KPC model (Aiello *et al*, 2016). The efficiency of PI3Kα targeting is not impacted by the heterogeneous genetic landscape of pancreatic cancer patients. A minimal PI3Kα activity is important for migratory ability of cells and contributes to their metastatic potential regardless of the genetic landscape of the tumour. In the model described by Thorpe *et al* (2015), PI3K isoform efficiency is for some patients predicted with the driving genetic alterations leading to PI3K hyperactivation: oncogenic Kras mutation promotes increased dependence to PI3Kα; loss of PTEN, increased dependence to PI3Kβ; oncogenic PI3Kα mutation, increased dependence to PI3Kα. We find that even *Pten*-depleted PDAC cell lines require a minimal PI3Kα activity to migrate. We support this result by demonstrating that a basal level of PI3Kα activation is sufficient to drive these pro-metastatic features (Fig 4). This result fits with the rare detection of oncogenic mutations in PI3Kα encoding gene (Waddell *et al*, 2015) despite the crucial role of PI3Kα activity in this cancer initiation (Baer *et al*, 2014; Wu *et al*, 2014). PI3Kα importance in controlling PDAC cell migration can occur independently from Kras mutation and is also promoted by FGFR signalling. There is a vast literature showing that FGFR2 is promoting PDAC cell migration *in vitro*, *in vivo* and in patients (Nomura *et al*, 2008). We could speculate that remodelling the actin cytoskeleton appears to be key in PDAC progression so that tumour cells can extrude themselves from the strong desmoplastic and low vascularised pancreatic primary tumours. This could also be the case for the evolution of micro-metastasis: others found that, in metastatic sites, macrophage VCAM binding critically activates PI3K-Ezrin in breast metastatic cells and promotes transition to macro-metastatic foci (Chen *et al*, 2011). Ezrin plays a critical role in maintaining cell shape and lamellipodial extensions (Lamb *et al*, 1997). Early on in the discovery of oncogenic KRAS properties, it was demonstrated that the PI3K signal towards actin remodelling was the first major signalling event leading to cell transformation (Rodriguez-Viciana *et al*, 1997). Subsequently, actin remodelling was also linked to PI3Kα-driven

glucose metabolism regulation (Hu et al, 2016). Actin cytoskeleton remodelling and associated cellular functions such as migration and motility are sensitive to unique and isoform-selective PI3Kα inhibition. Our data describing production of PIP₃ with distinctive acylation pattern points to a novel mechanism to explain PI3K isoform selectivity. Currently, published data are increasingly showing that the acylation state of phospholipids could modulate their localisation and function (Choy et al, 2017). In this context, we can provide additional information to show that pan-PI3K inhibitors or isoform-specific inhibitors could target different PIP₃ subspecies and conversion to PI-3,4-P₂ ultimately leading to clear-cut effects on migration or cytotoxicity and explaining isoform selectivity.

Thirdly, pro-tumoural macrophages in the organ around the tumour could favour metastasis evolution. Others found that activation of macrophages associated with increased apoptotic tumour cells accelerated growth and not implantation of prostate metastatic nodules (Roca et al, 2018). Similarly, systemic secretions of the tumour-associated macrophages that are found around the primary pancreatic tumours could favour growth of metastatic micro-metastatic foci. Tumour-intrinsic PI3Kα indirectly rewires immune cell composition in the tumour site; this increase of CD206-positive cell counts in the tissue corresponding to tumour-associated inflammatory macrophages could act as a distant event and promote metastatic evolution. IL-3 regulates varied inflammatory responses that promote the rapid clearance of pathogens but also contribute to pathology in chronic inflammation. Therapeutic interventions manipulating this cytokine are so far only developed in AML (Dougan et al, 2019), but increased understanding of its action in solid tumours is needed before their intervention being included to the arsenal of immunotherapies. Suppression of macrophages in the KPC mice and their primary tumours by clodronate treatment did not delay lethality, but decreased incidence of macro-metastasis (Griesmann et al, 2017). The tumour-intrinsic action of PI3Kα on tumour inflammation that we described here could be added to its direct immunomodulatory action in pancreatic cancer found by others (Sivaram et al, 2019). Hence, PI3Kα pro-metastatic action acts both via a direct actin cytoskeleton remodelling and FGFR-induced tumour cell migration as well as via tumour-extrinsic promotion of TNFα secretion by macrophages and subsequent further activation of migratory phenotype. These data are in line with our early work (Guillermet et al, 2003; Bousquet et al, 2006) that showed that class IA PI3K activity in pancreatic cancer cells is critical to activate NF-κB activity, preventing TNFα-induced cell death and promoting cell survival and migration. Finally, Fig 1A showed that Hallmark TNFα signalling via NFKB is one of the 6 hallmarks significantly changed in metastatic PDAC patients, further validating our finding. TNFα was previously found to promote transition from micro- to macro-metastasis as BM-derived EPCs are known to be critical regulators of the angiogenic switch in progression of micro-metastasis to lethal macro-metastasis (Gao et al, 2008), and tumour-derived TNF signalling had been linked in vivo to differentiation of myeloid progenitor cells to "byphenotypic" myeloid/ECs (Li et al, 2009).

Fourthly, the repartition of each KRAS mutation is different in each solid cancer, with G12D mutation being the most common in PDAC. Recent evidence shows that each KRAS mutation drives different signalling and engages different pathways (Cayron & Guillermet-Guibert, 2020; Hobbs et al, 2020). The published data

favour the prominent role of PI3Kα downstream KRASG12D in PDAC. However, the multiple PI3K isoform engagement could also explain why some tumours appear to escape from BYL-719 treatment in vivo. Interestingly, while all BYL-719-treated tumours presented a significant decrease in CD206 staining, the effects on CD4 and CD8 immune cell population were found heterogeneous; this is associated with the fact that some BYL-719-treated mice presented (albeit in a reduced size and number) macro-metastatic foci. Anti-tumoural action of PI3Kα partial genetic inactivation was also found heterogeneous in C57/B6 background. However, in the nude mice model, that is devoid of lymphocytes but presents macrophages, three mice out of eight displayed lower effects of BYL-719 treatment, suggesting that other parameters than immune cell rewiring could be involved in resistance to treatment. In vitro, some cell lines were also sensitive to other PI3K isoform inhibitors (e.g. PI3Kβ). Both PI3Kγ and PI3Kα are important for pancreatic cancer (Torres et al, 2019); the possible crosstalk between these two isoforms should be investigated. In a future study, we aim to dissect the mechanisms of resistance to PI3Kα inhibition, in an aim to increase the efficiency of this therapeutic agent on PDAC evolution.

In conclusion, our data demonstrate that PI3K-targeting agents could be effective in the management of micro-metastatic disease (assessed by cfDNA), in PDAC patients, preventing macro-metastatic evolution. PI3K inhibitors also trigger indirect immunomodulatory actions. Our work provides sufficient ground to support the emergence of an extended translational study in PDAC patients to corroborate these findings.

# Materials and Methods

### Transcriptomics and bioinformatics analysis

We selected transcriptional profiling data sets of normal pancreas, chronic pancreatitis and pancreatic primary tumoural tissues from localised (PDACloc) or metastatic patients (PDACmet). Published data on human samples were retrieved from public databases E_EMBL_6 (Abdollahi et al, 2007) from compatible platforms, normalised using the RMA method (R 3.2.3, bioconductor version 3.2), collapsed (collapse microarray), filtered (SD > 0.25) and statistically tested using an ANOVA test corrected using the Benjamini and Hochberg method (BH). For each sample, individual scoring for hallmarks or Reactome (actualised list of genes downloaded from MSigDB version [software.broadinstitute. org/gsea/msigdb] and Reactome [www.reactome.org]) was performed using Autocompare_SES software (available at https://sites.google.com/site/fred softwares/products/autocompare_ses) using the "greater" (indicating an enriched gene set) Wilcoxon tests with frequency-corrected null hypotheses (Tosolini et al, 2016), followed by values in each group of patients compared using an ANOVA test. Hierarchical patient clustering was performed using the PI3Kα activation signature. The PI3Kα gene signature was designed as the intersection of genes up-regulated and down-regulated in 20 breast tumours after BYL-719 treatment (Bosch et al, 2015) and LINCS shRNA CMAP sig gene list (Zhang et al, 2017) (Appendix Fig S1). This list was narrowed down to 20 genes, which expression was found compatible with the quality criteria (filter) detailed above. Unsupervised hierarchical clustering of the E_EMBL_6 data set was performed

focusing on the expression of these 20 genes regulated by PI3Kα in cancer.

Confirmed PDAC samples from public databases were selected for further bioinformatics analysis. In detail, mRNA expression data and clinical data from confirmed PDAC patients of PAAD (TCGA) (175 patients) and PACA-AU (267 patients) cohorts were retrieved. Amongst the 175 well-annotated TGCA patients, 21 patients were considered localised according to their UICC staging (T = 0, 1 or 2, N = 0, M = 0). For each patient, a PI3Kα activation signature score or immune cell infiltration score (to quantify LTγδ, NK, LT CD8, Monocyte-Macrophage-DC, B cells, granulocytes, LT CD4) was given using SES auto compare software, and patients were hierarchically clustered in three groups corresponding to high, medium or low scoring. Scores in each group were statistically tested using an ANOVA test corrected according to the Benjamini and Hochberg method (BH). For PI3Kα activation signature, high and medium groups were then pooled. The overall survival of patients in each cluster was plotted and statistical differences were calculated using the log rank test. The prognosticator value of PI3Kα activation signature and IUCC staging were tested independently and compared to the value of clinical T,N,M staging using the multivariate Cox test and the PAAD database. We verified that within each patient cluster, there was no enrichment in terms of genetic changes associated with the PI3K/Akt pathway and, in particular, that oncogenic mutations of PI3Kα and PTEN were infrequent and equally distributed in each group of patients (mutational pattern available in PAAD cohort only: Dataset EV3, Appendix Fig S2). SES scores for PI3Kα activation were also calculated in PDAC RNA subtypes using classification method from Puleo *et al* (2018).

**Human pancreatic samples**

Patient samples from BACAP collection were collected and stored with the "CRB Cancer des Hôpitaux de Toulouse". Informed consent was obtained from all subjects and the experiments conformed to the principles set out in the WMA Declaration of Helsinki and the Department of Health and Human Services Belmont Report. The BACAP collection has been declared to the Ministry of Higher Education and Research (DC-2008-463), and transfer agreements (AC-2008-820) (AC-2013-1955) have been obtained following approval by ethical Committees. Clinical and biological annotations of the samples have been declared to the CNIL (Comité National Informatique et Libertés—French national Data Protection Agency). All patient records and information were anonymised and encrypted prior to analysis. IHC is detailed in Appendix, antibody used in Dataset EV4.

Blood was collected from patients diagnosed with pancreatic adenocarcinoma and analysed by digital droplet PCR (ddPCR) for KRAS mutation. Briefly, 10 ml of blood was centrifuged twice in PAXgene blood ccfDNA tubes (Qiagen) at 1,200 *g* for 10 min at 4°C and 16,000 *g* for 10 min at 4°C. Cell-free plasma was collected and total DNA was extracted from 3ml of plasma using the QIAamp Circulating Nucleic Acid Kit (Qiagen) according to the manufacturer's recommendation. Circulating cell-free DNA (cfDNA) ranging from 110-210 base pairs was qualified and quantified using the DNF-474 high sensitivity ngs fragment analysis kit (Agilent). For ddPCR, 5 ng of cfDNA were analysed using the ddPCR™ KRAS G12/ G13 Screening Kit #1863506 (Bio-Rad) and the QX200 Droplet

Digital PCR (ddPCR) System (Bio-Rad), according to the manufacturer's recommendation. Samples with high-molecular DNA were excluded from the analysis. Allelic frequency percentages for KRAS mutation were obtained using QuantoSoft software (Bio-Rad).

**Inhibitors and ligands**

For *in vitro* use, all PI3K inhibitors (Knight *et al,* 2004; Jackson *et al,* 2005; Pomel *et al,* 2006; Ali *et al,* 2008; Folkes *et al,* 2008; Raynaud *et al,* 2009; Burger *et al,* 2011; Jamieson *et al,* 2011; Fritsch *et al,* 2014; Barlaam *et al,* 2015) (Dataset EV4) and the FGFR inhibitor AZD4547 were purchased (CliniSciences) and dissolved in dimethyl sulfoxide (DMSO) to obtain a stock concentration of 10 mM, subsequently diluted as indicated and compared to the diluted DMSO vehicle (vehicle). *In vivo*, BYL-719 (ApexBio) was dissolved in 0.5% methyl cellulose with 0.2% Tween-80 and administered by oral gavage at 50 mg/kg daily. TNF-alpha was purchased (Sigma-Aldrich) and dissolve in PBS to obtain a stock concentration of 1 mg/ml.

**Cell lines**

All the cell lines described in Dataset EV4 were obtained from the American Type Culture Collection (ATCC, Manassas, VA) or from genetically engineered mouse models or from CRB, Toulouse, IUCT-O. Mutant KRAS, with or without partially deficient PI3Kα activity pancreatic cancer cell lines were derived from the pancreas and lung of KC or KC;p110α$^{+/lox}$ animals (aged 10–13 months). Peritoneal metastatic cells pASC1 and pASC3 are primary cells isolated from metastatic patients with PDAC in the IUCT-O. R211-Luc cells are R211 cells modified to express the luciferase. General methods were used and are detailed in the Appendix and Dataset EV4. IC21 cells were treated for 1 day with the conditioned medium, pre-incubated with control Antibody (0.5 μg/ml, MAb rat IgG1 isotype control) or anti-IL3 antibody (0.5 μg/ml, Clone MP2-8F8, Catalog number: BX-BE0282-1MG InVivoMab anti-mouse IL-3) and TNFα measured from supernatant. Lipids were extracted and derivatised using TMS-diazo-methane as previously described (Clark *et al,* 2011).

**siRNA, stably transfected cell lines**

Cells were transfected with Lipofectamine 2000 (ThermoFisher Scientific) with SMARTpool ON-TARGETplus mouse siRNA (Dharmacon) targeting: Pik3ca, Pik3cb, Pik3cg, Pik3cd according to the manufacturer's protocols as used in (Kingham & Welham, 2009; Höland *et al,* 2014; Huang *et al,* 2015). ON-TARGETplus Non-targeting control siRNAs (Dharmacon) were used as controls. Twenty-four hours after transfection, the cells were used for migration or RT–qPCR experiments. Panc-1 cells were stably transfected by lentiviral transduction with pLVTHM-p110alpha1 (forward: 5′CGCGTCCCCGCGAAATTCTCACACTATTATTTCAAGAGAATAAT AGTGTGAGAATTTCGCTTTTTGGAAAT; reverse: 5′CGATTTCCAAA AAGCGAAATTCTCACACTATTATTCTCTTGAAATAATAGTGTGAG AATTTCGCGGGGA); pLVTHM-p110alpha2 (forward: 5′CGCGTCC CCGCACAATCCATGAACAGCATTTTCAAGAGAAATGCTGTTCATG GATTGTGCTTTTTGGAAAT; reverse: 5′CGATTTCCAAAAAGCACA ATCCATGAACAGCATTTCTCTTGAAATGCTGTTCATGGATTGTGC GGGGA), pLVTHM-p110beta1 (forward: 5′CGCGTCCCCCACATT

GACTTTGGACATATCGCGTCCCATATGTCCAAAGTCAATGTGGTTTT TGGAAAT; reverse: 5'CGATTTCCAAAAACCACATTGACTTTGGAC ATATTCTCTTGAAATATGTCCAAAGTCAATGTGGGGGGA), pLVTHM-p110beta2 (forward: 5'CGCGTCCCCCGACAAGACTGCCGAGAGATT TTCAAGAGAAATCTCTCGGCAGTCTTGTCGTTTTTGGAAAT; reverse: 5'CGATTTCCAAAAACGACAAGACTGCCGAGAGATTTCTCTTGAAA ATCTCTCGGCAGTCTTGTCGGGGGA), pLVTHM-scr (forward: 5'CG CGTCCCCTTCTAGAGATAGTCTGTACGTTTCAAGAGAACGTACAGA CTATCTCTAGAATTTTTGGAAAT; reverse: 5'CGATTTCCAAAAAT TCTAGAGATAGTCTGTACGTTCTCTTGAAACGTACAGACTATCTCT AGAAGGGGA) and selected positively by cell sorted flow cytometry for GFP expression.

## Animal models

All animal procedures were conducted in compliance with the Ethics Committee pursuant to European legislation translated into French Law as Décret 2013-118 dated 1st of February 2013 (APAFIS 3601-2015121622062840).

The LSL-KRASG12D (K) and LSL-p53R172H (P) knock-in from D. Tuveson, Mouse Models of Human Cancers Consortium Repository, Frederick National Cancer Institute, Pdx1-Cre (C) from D.A. Melton, Harvard University, Cambridge, MA, Pdx-1Cre (C) from D. Tuveson and p110αlox/lox from B. Vanhaesebroeck, University College London strains were interbred against a mixed background (CD1/SV129/C57Bl6) or in a C57/B6 background. KPC mice are of mixed gender. KPC mice and compound mice with p110αlox line were bred in three animal houses (mixed background, Melton's Pdx1-Cre: CRCT, Anexplo, Toulouse, France, 2 sites) (Therville *et al*, 2019) and (C57B6 background, Tuveson's Pdx1-cre: CRCM, Marseille, France). The Ptf1a^Cre/+-LSL-KRAS^G12D/+ (KC; p110α^+/+) and Ptf1a^Cre/+-LSL-PIK3CA^H1047R/+ (p110α^H1047R) strains were bred at the Technische Universität München (Technical University, Munich). All mice were housed and bred under specific pathogen-free conditions maintained in the accredited animal facility. Mice were housed with a 12-h day–night cycle with lights on at 7:00 AM in a temperature (22 ± 1°C) and humidity (55 ± 5%)-controlled room. All mice were allowed free access to water and food. All cages contained wood shavings, bedding and a cardboard, an igloo or a sizzle nest tube for environmental enrichment. KPC mice were treated 5 days a week with vehicle or BYL-719 administered by oral gavage at 50mg/kg daily starting the day after the tumour detection date.

## Tail vein injection

All obtained mice were acclimatised for at least 1 week. Vendor health reports indicated that the mice were free of known viral, bacterial and parasitic pathogens. $5 \times 10^4$ R211-Luc cells were injected into the tail vein of female 8-week-old nude mice (Charles River). The mice were treated 5 days a week with vehicle or BYL-719 administered by oral gavage at 50 mg/kg daily for 3 weeks starting on the injection date. Two- and three-week post-injection, mice were injected i.p. with 150 mg/kg of RediJect D-Luciferin (Perkin Elmer) and monitored for Luciferase expression after 6–8 min in the IVIS Spectrum *in vivo* imaging system (Perkin Elmer). Luminescence (photo count) was measured for each mouse. Lungs were inflated then fixed and embedded in paraffin. $1 \times 10^5$ A338 or $1 \times 10^5$ A260 cells were injected into the tail vein of female 8-week-old C57/B6

mice (Charles River). Three-week post-injection, lungs were inflated then fixed and embedded in paraffin.

## Pancreatic inflammation

Pancreatic injury was induced on young KC mice (8–12 weeks old) by a series of six hourly intra-peritoneal injections of caerulein (75 μg/kg of body weight, Bachem) repeated after 48 h. Eight days later, a treatment/intervention protocol was developed to allow formation of pre-cancer lesions: mice were treated with the vehicle (0.5% methyl cellulose with 0.2% Tween-80) or with GDC0326 (10 mg/kg) by gavage (Appendix Fig S16A).

## KPC cohort follow-up (ultrasound imaging, blood collection and cytokine profiling)

Ultrasound imaging was performed using the VisualSonics Vevo2100 High-Resolution System equipped with an ultrasound transducer in the 25–55 MHz range. Animal preparation and imaging procedures were performed as described in Sastra and Olive (2013). KPC mice were monitored once a week from 12 weeks old onwards; when a tumour was detected, ultrasound scans were performed every other day. The tumour area was measured by delimiting the tumour border and determining the major axis; at least five replicates were performed per mice per ultrasound. Tumour volume was calculated using the formula $V = (4/3) \times \pi \times (\text{Length}/2)^2 \times (\text{Depth}/2)$. The tumour volume fold change corresponds to the increased fold change in the tumour volume after treatment onset.

Blood samples were collected every 2 weeks via retro-orbital collection from the age of 12 weeks onwards. Blood counts were performed using Yumizen H500 haematology analyser (HORIBA), calibrated for murine blood. A maximum volume of 100–150 μl was collected on each occasion. The blood was collected using Pasteur glass pipettes and then transferred to Eppendorf tubes containing 20 μl 0.5 M EDTA. Plasma was separated by centrifuging blood at 1,500 *g* for 20 min at 4°C within 3 h of blood collection. The plasma was stored at −80°C until required for further use.

A MILLIPLEX®_MAP (Merc Millipore # MCYTMAG70PMX32BK) assay was performed using 25 μl of non-diluted murine blood plasma. We specifically tested for a panel of 32 murine chemokines and cytokines: Eotaxin, G-CSF, GM-CSF, IFNγ, IL-1α, IL-1β, IL-2, IL-3, IL-4, IL-5, IL-6, IL-7, IL-9, IL-10, IL-12 (p40), IL-12 (p70), IL-13, IL-15, IL-17, IP-10, KC, LIF, LIX, MCP-1, M-CSF, MIG, MIP-1α, MIP-1β, MIP-2, RANTES, TNFα and VEGF. IL3 and TNFα could not be detected on a comparative scale.

## cfDNA extraction from murine blood plasma

For cfDNA extraction, blood plasma was re-centrifuged at 18,000 *g* for 10 min at room temperature to reduce debris contamination. cfDNA was extracted from blood plasma using the QIAmp DNA Mini Kit (QIAGEN) protocol except for eluting the cfDNA in 50 μl of elution buffer. cfDNA samples were stored at −20°C until required for further use. CfDNA quantification was performed by qPCR as described below.

For quantifying the relative cfDNA in blood plasma, we initially plotted a standard curve using extracted DNA from a murine cell line R211 (without LSL cassette, expressing mutated *KRAS* and *TP53*)

and from pancreatic extracts from mice expressing the LSL cassette (not recombined). The qPCR was performed using 1 μL of DNA and SsoFast Eva Green Supermix (Bio-Rad®). For the standard curve, we did serial dilutions up to 1/1,000 of the two DNA extracts and a qPCR of two different genes, p53 and GAPDH. We designed and selected the most specific and efficient primers (Sigma-Aldrich®) using Primer-Blast (NCBI). The obtained standard curves allowed subsequent quantification of cfDNA in mouse plasma samples.

The following primers were used: p53 gene (Forward: 5′-CC AGCTCAGCCTTTGTAGTGAA-3′; reverse: 5′-GTGCAGCCCTAAGCA TCTAGC-3′; chromosome 11), GAPDH gene (Forward: 5′-AGCCC CAGGCTATCTGATGT-3′; 5′-ATAGCTGATGGCTGCAGGT-3′; chromosome 6).

cfDNA thresholds were calculated by compiling all cfDNA measurements from mice with normal pancreas, high-grade PanINs, localised and metastatic PDAC (Dataset EV5). For the survival curve (Kaplan–Meier), mice were categorised as having a normal, low or high level of cfDNA according to their mean cfDNA values. As for thresholds, the low range corresponds to mice presenting 0 – 0.003 AU of cfDNA and the high range to mice presenting cfDNA above the mentioned level. In order to establish these ranges, we analysed the histology of each mouse at end point. cfDNA was quantified using blood collected at sacrifice/death. All raw data and threshold calculations are shown in Dataset EV5.

The Fragment Analyzer™ was used to determine the size of DNA fragments in blood plasma. The DNF-474 High Sensitivity NGS Fragment Kit was used to characterise cfDNA. Three different cfDNA fragmentation profiles were obtained: for normal and healthy mice, the electropherogram did not present any fragment; for mice with high-grade PanINs and localised PDAC, a 160–210 bp fragment was always found; for mice with metastatic PDAC, the electropherogram presented the aforementioned 160–210 bp fragment, in addition to larger fragments but at lower concentrations.

## Statistical analysis

Experimental data provided at least three experimental replicates and three to five measurement replicates, except for WB experiments. Statistical analyses were performed with GraphPad Prism using student *t*-test for *in vitro* data; non-parametric Mann–Whitney for all *in vivo* data; Log rank test was used for all survival curves: $*P < 0.05$, $**P < 0.01$, $***P < 0.001$. Correlation analysis was performed using Pearson *r* test. Statistical relevance of the murine KPC cohort size was determined using a Power Calculation test (www.lasec.cuhk.edu.hk/.../power_calculator_14_may_2014.xls), those data are shown in Dataset EV5. For patient data analysis, ANOVA test corrected using the Benjamini and Hochberg method (BH) on SES (sample enrichment score).

# Data availability

This study includes no data deposited in external repositories. Normalised raw data of analysed public data are available in Dataset EV1 enabling a further analysis on the built cohorts (normal, PDAC loc, PDAC met, CP).

**Expanded View** for this article is available online.

## The paper explained

### Problem

Pancreatic cancer is one of the most lethal solid cancers and is characterised by rapid progression after primary tumour detection. The key signalling events driving this fast evolution into macro-metastatic disease are still unknown.

### Results

Two unbiased approaches led to the identification of a high PI3Kα activation signature in pancreatic primary tumours with bad prognosis. Our *in vitro* data showed that PI3Kα is a major positive regulator of cancer cell escape from the primary tumour through actin cytoskeleton remodelling. PI3Kα was inhibited in two preclinical models of pancreatic cancer. First, in a model of micro-metastatic disease (extra-pancreatic dissemination that goes undetected by ultrasound (US) imaging), mice presenting US-detected primary pancreatic tumours and increased circulating cell-free DNA (cfDNA) were treated. In a second model, tumour cell implantation and early proliferation in metastatic organs after intravascular injection were analysed. A clinically relevant PI3Kα-selective inhibitor (BYL-719/Alpelisib), currently tested in pancreatic cancer patients without patient stratification, was used in both models. Inhibition of PI3Kα delayed primary tumour and micro-metastasis evolution, showing that PI3Kα activity drove the evolution of micro-metastatic disease towards the macro-metastatic stage in these models. Mechanistically, tumour-intrinsic PI3Kα activity increased pro-tumoural characteristics in peritumoural immune cells via increased IL-3 cytokine production. In return, inflammatory macrophages increased TNFα production, facilitating tumour cell migration.

### Impact

In pancreatic cancer patients, PI3Kα-targeting agents could be effective in the management of micro-metastatic disease assessed by cfDNA, preventing macro-metastatic evolution. PI3Kα inhibitors also trigger indirect immunomodulatory actions and could be added in the arsenal of immunomodulatory agents.

# Acknowledgements

We are grateful to SigDYN members, past and present, for their technical support, sample banks, common tools, scientific and protocol discussions, CRB and BACAP consortium for patient samples, UMS006/CREFRE, Anexplo Platform, Toulouse (mouse breeding and experimental zone; ENI core platform), the CRCT core technology platform in particular Laetitia Ligat, Carine Valle and Emeline Sarot, ImagIN platform (FX Fresnois), Dr. Barbara Garmy-Susini for access to her laboratory while moving our own facilities, the staff of Histology platform (I2MC, Inserm 1048, Toulouse, France), MetaToul-Lipidomique Core Facility (I2MC, Inserm 1048, Toulouse, France), MetaboHUB-ANR-11-INBS-0010 for lipidomic analysis, advices and technical assistance, and Genentech for GDC0326. JGG is a member of COST action EU Pancreas BM1204. JGG's laboratory belongs to Toucan, Laboratoire d'Excellence, ANR, an integrated research programme on Signal-targeted Drug Resistance. JGG's laboratory for this topic was/is funded by Europe EU-ERG FP7 (270696 PaCa/PI3K), ARC (PJA20171206596; salary for RB), Toucan ANR Laboratory of Excellence, MSCA-ITN/ETN PhD-PI3K (Project ID: 675392, salary for FR-D and SA), Fondation de France (salary for BT), GSO, Ligue Nationale Contre le Cancer (salary for CC and CC), Fondation Toulouse Cancer Santé (revision experiments). Both FR-D and SA disseminated their research to high school students, as part of their commitment in MSCA-ITN funding. DS laboratory for this research was funded by Deutsche Forschungsgemeinschaft (DFG, German Research Foundation)—Project ID 329628492—SFB 1321 (DS) and the European Research Council (ERC CoG No. 648521 (D.S.).

## Author contributions

BT, FR-D, EP-T, NT, CC, SA, SC-S, GR-G, MT, AVV, CC, RB, JB-M, DP, DFM, AC, JG-G: experiments; BT, FR-D, EPT, SA, GRC, MT, AVV, CC, AC, PC, CB, JG-G: formal analysis of data; BT, FR-D, EP-T, NT, CC, EA, PC, DS, CB, JG-G: methodology; BT, FR-D, EP-T, MT, JG-G: visualisation of data; BT, FR-D, JG-G: writing—original draft; BT, FR-D, EP-T, NT, MT, JB-M, PC, JG-G: materials and methods writing; BT, FR-D, EP-T, JG-G: review of writing; all authors: editing MS; GRC, HY, CF, FM, BB, AC, DS, JG-G: provided samples; J-PD, EA, AC, PC, DS, CB, JG-G: supervision; BT, FR-D, EP-T, JB-M, DS, JG-G: funding acquisition; JG-G: conceptualisation, project administration, project supervision, validation.

## Conflict of interest

EPT was funded by Cellgene. WO2021001431-A1 is a filed patents pertaining to the results presented in the paper.

## For more information

i   https://www.youtube.com/watch?v=GE_A-ApZ-cA

ii  https://www.youtube.com/watch?v=WbFmrM7m7Yk

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
