## [Review Process File · EMBO Molecular Medicine]

Pancreatic Cancer Intrinsic PI3Ka Activity accelerates Metastasis and rewires Macrophage Component

Benoit Thibault, Fernanda Ramos-Delgado, Elvire Pons-Tostivint, Nicole Therville, Celia Cintas, Silvia Arcucci, Cassant-Sourdy Stéphanie, Gabriela Reyes-Castellanos, Marie Tosolini, Amelie Villard, Coralie Cayron, Romain Baer, Justine Bertrand-Michel, Delphine Pagan, Dina Ferreira Da Mota, Hongkai Yan, Chiara Falcomata, Fabrice Muscari, Barbara Bournet, Jean-Pierre Delord, Ezra Aksoy, Alice Carrier, Pierre Cordelier, Dieter Saur, Celine Basset, and Julie Guillermet-Guibert
DOI: 10.15252/emmm.202013502

Corresponding author: Julie Guillermet-Guibert (julie.guillermet@inserm.fr)

Review Timeline:

Submission Date:	23rd Sep 20
Editorial Decision:	12th Oct 20
Revision Received:	12th Mar 21
Editorial Decision:	30th Mar 21
Revision Received:	17th Apr 21
Accepted:	23rd Apr 21

Editor: Lise Roth

Transaction Report:

12th Oct 2020

Dear Dr. Guillermet-Guibert,

Thank you for submitting your work to EMBO Molecular Medicine. We have now heard back from the two referees who agreed to evaluate your manuscript. As you will see below, the reviewers raise substantial concerns on your work, which preclude its publication in EMBO Molecular Medicine in its current form.

However, the referees agreed that despite a number of serious concerns, the manuscript should be given a chance, and we will therefore welcome a resubmission of your manuscript pending satisfactory major revisions.

Addressing the reviewers concerns in full will be necessary for further considering the manuscript in our journal. Still, revising the manuscript according to the referees' recommendations appears to require a lot of additional work and experimentation. I am unsure whether you will be able or willing to address those and return a revised manuscript within the 3-6 months deadline. On the other hand, given the potential interest of the findings, I would be willing to consider a revised manuscript with the understanding that acceptance of the manuscript would entail a second round of review. EMBO Molecular Medicine encourages a single round of revision only and therefore, acceptance or rejection of the manuscript will depend on the completeness of your responses included in the next, final version of the manuscript. Should you find that the requested revisions are not feasible within the constraints outlined here and prefer, therefore, to submit your paper elsewhere, we would welcome a message to this effect.

When submitting your revised manuscript, please carefully review the instructions that follow below. Failure to include requested items will delay the evaluation of your revision:

- 1) A .docx formatted version of the manuscript text (including legends for main figures, EV figures and tables). Please make sure that the changes are highlighted to be clearly visible.
- 2) Individual production quality figure files as .eps, .tif, .jpg (one file per figure).
- 3) A .docx formatted letter INCLUDING the reviewers' reports and your detailed point-by-point responses to their comments. As part of the EMBO Press transparent editorial process, the point-by-point response is part of the Review Process File (RPF), which will be published alongside your paper.
- 4) A complete author checklist, which you can download from our author guidelines (<https://www.embopress.org/page/journal/17574684/authorguide#submissionofrevisions>). Please insert information in the checklist that is also reflected in the manuscript. The completed author checklist will also be part of the RPF.
- 5) Before submitting your revision, primary datasets produced in this study need to be deposited in an appropriate public database (see <https://www.embopress.org/page/journal/17574684/authorguide#dataavailability>).

Please remember to provide a reviewer password if the datasets are not yet public. The accession numbers and database should be listed in a formal "Data Availability " section (placed after Materials & Method). Please note that the Data Availability Section is restricted to new primary data that are part of this study.

6) We would also encourage you to include the source data for figure panels that show essential data. Numerical data should be provided as individual .xls or .csv files (including a tab describing the data). For blots or microscopy, uncropped images should be submitted (using a zip archive if multiple images need to be supplied for one panel). Additional information on source data and instruction on how to label the files are available at .

7) Our journal encourages inclusion of *data citations in the reference list* to directly cite datasets that were re-used and obtained from public databases. Data citations in the article text are distinct from normal bibliographical citations and should directly link to the database records from which the data can be accessed. In the main text, data citations are formatted as follows: "Data ref: Smith et al, 2001" or "Data ref: NCBI Sequence Read Archive PRJNA342805, 2017". In the Reference list, data citations must be labeled with "[DATASET]". A data reference must provide the database name, accession number/identifiers and a resolvable link to the landing page from which the data can be accessed at the end of the reference. Further instructions are available at .

8) We replaced Supplementary Information with Expanded View (EV) Figures and Tables that are collapsible/expandable online. A maximum of 5 EV Figures can be typeset. EV Figures should be cited as 'Figure EV1, Figure EV2" etc... in the text and their respective legends should be included in the main text after the legends of regular figures.

- Additional Tables/Datasets should be labeled and referred to as Table EV1, Dataset EV1, etc. Legends have to be provided in a separate tab in case of .xls files. Alternatively, the legend can be supplied as a separate text file (README) and zipped together with the Table/Dataset file. See detailed instructions here: .

9) For more information: There is space at the end of each article to list relevant web links for further consultation by our readers. Could you identify some relevant ones and provide such information as well? Some examples are patient associations, relevant databases, OMIM/proteins/genes links, author's websites, etc...

10) Author contributions: the contribution of every author must be detailed in a separate section (before the acknowledgments).

11) Every published paper now includes a 'Synopsis' to further enhance discoverability. Synopses are displayed on the journal webpage and are freely accessible to all readers. They include a short stand first (maximum of 300 characters, including space) as well as 2-5 one-sentences bullet points

that summarizes the paper. Please write the bullet points to summarize the key NEW findings. They should be designed to be complementary to the abstract - i.e. not repeat the same text. We encourage inclusion of key acronyms and quantitative information (maximum of 30 words / bullet point). Please use the passive voice. Please attach these in a separate file or send them by email, we will incorporate them accordingly.

Please also suggest a striking image or visual abstract to illustrate your article. If you do please provide a png file 550 px-wide x 400-px high.

12) As part of the EMBO Publications transparent editorial process initiative (see our Editorial at <http://embomolmed.embopress.org/content/2/9/329>), EMBO Molecular Medicine will publish online a Review Process File (RPF) to accompany accepted manuscripts.

In the event of acceptance, this file will be published in conjunction with your paper and will include the anonymous referee reports, your point-by-point response and all pertinent correspondence relating to the manuscript. Let us know whether you agree with the publication of the RPF and as here, if you want to remove or not any figures from it prior to publication.

I look forward to receiving your revised manuscript.

Yours sincerely,

Lise Roth

Lise Roth, PhD
Editor
EMBO Molecular Medicine

To submit your manuscript, please follow this link:

Link Not Available

Each figure should be given in a separate file and should have the following resolution:
Graphs 800-1,200 DPI

Photos 400-800 DPI

Figures are not edited by the production team. All lettering should be the same size and style; figure panels should be indicated by capital letters (A, B, C etc). Gridlines are not allowed except for log plots.

Figures should be numbered in the order of their appearance in the text with Arabic numerals. Each Figure must have a separate legend and a caption is needed for each panel.

***** Reviewer's comments *****

Referee #1 (Remarks for Author):

This is very interesting study providing evidence for "personalized" targeting of PI3K in advance cases of pancreatic cancer. The study is well executed and data clearly presented. However, additional experimentation is needed to fully support the authors conclusions and enhance the translational significance of the manuscript. First, the gene expression analysis of Figure 1 should be validated in an independent cohort different than the dataset from publicly available sources. More importantly, it is essential to validate that gene signature is associated with the activation of PI3K using the well validate substrates. Also, the levels of circulating DNA should be included in a larger cohort of cases. This will provide solid evidence to support future clinical studies in the personalized medicine setting. Second, the authors should define if there is an association of the PI3K signature and the major PDAC RNA subtypes. Third, the in vivo tail vein injection experiment should be validated using genetic tools in a syngeneic model. Fourth, the upstream signaling should be evaluated. The connection with FGFR signaling is quite interesting; is PI3K the main mediator of the pro-migratorial effect of FGFR? Finally, the last section on the macrophages is a bit out place, I suggest to take it out.

Referee #2 (Comments on Novelty/Model System for Author):

Mouse model only very partially recapitulate the complexity of human PDACs

Referee #2 (Remarks for Author):

In this study Thibault et al propose a role for PI3K alpha in the control of PDAC cell migration, tumor progression and polarization of tumor-associate macrophages.

The rationale for the study is clear and some conclusions coherent with the experimental findings. However, the study presents several important weaknesses.

1. The differential enrichment of one specific PI3K-related pathway (Fig 1a,b) in metastatic vs. localized PDAC samples likely reflects the presence in this specific signatures of genes that are not present in the other two related signatures in Fig. 1b. This should be investigated and explained.
2. The links between the PI3K pathway and cell migration is far from being novel as there is a very extensive and mature literature in this area. In addition, since the PI3K pathway is downstream of activated KRAS and thus has an obvious impact on cell viability and proliferation, it is essential that the effects of these inhibitors on viability are carefully measured with assays more sensitive and specific than the MTT assay used here (e.g. in Fig 2f), such as Annexin V staining or equivalent.

Finally, in migration assays it would be important to understand what are the properties of the cell lines analyzed. PANC1 have quasi-mesenchymal features and they are very different from other human PDAC cell lines. However, the features of the mouse cell lines used here and to what extent they have mesenchymal vs epithelial properties are unclear.

3. The interpretation of the links between PI3K, metastatization and detection of circulating tumor DNA is rather questionable. The amount of circulating tumor DNA correlates with the overall size of the tumor (including the primary tumor and metastases, if present), not necessarily with the metastatic behavior. In Fig. 5 the authors are comparing pT1N0 tumors with pT2N1 tumors, namely tumors with an overall low mass vs. tumors with an overall high mass. The higher level of ctDNA in the latter group is simply a reflection of this higher tumor mass, not of the different metastatic properties of these two tumors.

4. Links between PI3K and immune system. The difference in blood cell counts are of difficult interpretation in the context of metastatic KPC mice as they may represent indirect consequences of an advanced disease (with tumor necrosis etc.). More importantly, how the differences in IL3 expression would be linked to differences in stromal macrophages is completely unclear.

Dear Editor,

We thank you for the opportunity to revise our MS EMM-2020-13502 entitled "Pancreatic Cancer Intrinsic PI3K α Activity accelerates Metastasis and rewires Macrophage Component". Please find enclosed the response to reviewers's comments.

In the revised manuscript, we addressed the Reviewer's critiques and have modified the manuscript accordingly. A point-by-point response is detailed below, and the modified text is highlighted in yellow in the manuscript.

The Reviewer's comments were extremely helpful, and we appreciate such constructive feedback on our initial submission. After addressing their concerns, we are convinced that the quality of the paper has substantially improved, and we hope that the changes were made in accordance with their requirements. Should you require any further information and/or clarification, we would gladly provide it.

Yours sincerely,

Dr. Julie Guillermet-Guibert

Referee #1 (Remarks for Author):

This is very interesting study providing evidence for "personalized" targeting of PI3K in advance cases of pancreatic cancer. The study is well executed and data clearly presented. However, additional experimentation is needed to fully support the authors conclusions and enhance the translational significance of the manuscript.

1.1- First, the gene expression analysis of Figure 1 should be validated in an independent cohort different than the dataset from publicly available sources. More importantly, it is essential to validate that gene signature is associated with the activation of PI3K using the well validate substrates. Also, the levels of circulating DNA should be included in a larger cohort of cases. This will provide solid evidence to support future clinical studies in the personalized medicine setting.

We thank the reviewer for these questions. We first confirmed that cancer cells isolated from peritoneal metastasis (two primary culture of patient-derived cells) presented a significantly higher expression of *FOXA1*, *RRM2*, *BIRC5*, *FGFR4*, *PHGDH*, *TYMS*, as well as *MYBL2*, *PTTG1*, *KIF2C*, *CDC20*, *CCNB1* albeit in a lower extent compared to non-tumoural ductal cells (New Suppl. Fig. 1b). Indeed, this gene sub-set of the PI3K α activation signature was increased in all metastatic patients (Fig. 1c); those cells also presented increased levels of pS473Akt/Akt levels compared to non-tumoural ductal cells (New Suppl. Fig. 1c) analysed by WB. β -actin was used as a loading control.

With regards to the levels of circulating DNA, previously performed on two patients, we extended the study to 6 patients with similar T3 staging; this later restriction to T3 staging allow us to take into account Reviewer2's comment. In T3N0M0 patients, level of pAKT substrate was low; low cfDNA T3N1M0 patients presented lower pAKT substrate levels than high cfDNA T3N1M0 patients (New Fig. 5a,b). (Of note, we divided Figure 5 in two figures to help the reading of the figures.)

The results in Fig. 1 were performed on three independent cohorts of patients. To further validate our data, we tested the predictive value of PI3K α activation signature in two other cohorts that were more recently published as shown in Fig. Rev1. High scoring of PI3K α activation was also significantly increased in patients with the poorest prognosis, regardless of their stage.

Fig. Rev1. A PI3K α activity specific transcriptomic signature predicts pancreatic cancer aggressiveness. Scoring of the PI3K α activation transcriptomic signature was used to cluster patients with high (red) and low (green (a) or black (b)) scoring levels in the primary tumours of confirmed PDAC patients from E_MTAB_6134, Puleo *et al* or in PDX of PDAC patient from E_MTAB_5039 Nicolle *et al* (BH corrected p-values). The survival curves of each cluster were then plotted, and the statistical significance was calculated using the logrank test.

We agree with the reviewer and are convinced that, together with all the results gathered in the manuscript, these data provide sufficient ground to support the emergence of an extended translational study in PDAC patients; this is however out of scope of the current work and will be performed in a second step of our project by a prospective analysis. It will include quantification of pAktSubstrate, inflammatory macrophage infiltration in primary tumour by IHC, of cfDNA and circulating cytokines levels in plasma (IL3/TNF α) as well as PI3K α activation transcriptomics signature.

1.2-Second, the authors should define if there is an association of the PI3K signature and the major PDAC RNA subtypes.

We thank the reviewer for this suggestion and used the classifiers of Puleo *et al* (PMID: 30165049). In both cohortes, PI3K α activation signature is mostly associated with pure basal like RNA subtype as described in Suppl. New Fig2b.

We discussed this finding as follows: “Interestingly, patients with PI3K α activation signature are also enriched in pure basal-like phenotype. This RNA-based PDAC subtype is known as the most aggressive subtype of PDAC patients (PMID: 30165049, PMID: 21460848). Mueller *et al.*, (PMID: 29364867) showed that this subtype of murine cell lines showed a strong gene set enrichment for Ras downstream signaling pathways, including PI3K/Akt signalling, further corroborating our finding.”

1.3-Third, the in vivo tail vein injection experiment should be validated using genetic tools in a syngeneic model.

Similar results were observed in C57/B6 mice when comparing area of metastasis foci after tail vein injection of syngeneic Kras mutant pancreatic cells (A338) compared to Kras mutant pancreatic cells partly lacking PI3K α activity through a genetic inactivation of one allele of *Pik3ca* (A260, see also M&M) (New Fig. 6n).

1.4-Fourth, the upstream signaling should be evaluated. The connection with FGFR signaling is quite interesting; is PI3K the main mediator of the pro-migratorial effect of FGFR?

We thank the reviewer for this suggestion. FGFR signal activation was prevented by treatment with 2 μ M of AZD4547. AZD4547 significantly decreased R211 cell migration (New Fig. 2f). The concomitant inhibition of FGFR and PI3K α by simultaneous BYL-719 and AZD4547 treatment does not inhibit more cell migration than individual treatment which suggests that at least a part of PI3K α pro-migratory signal is due to FGFR activation.

1.5-Finally, the last section on the macrophages is a bit out of place, I suggest to take it out.

We thank the reviewer for pointing this out and have improved the data related to this Figure. Transition from micro- to macro-metastasis is now well described as promoted by tumour-extrinsic factors including by immune cells (PMID: 27083997). We hence analysed systemic alteration of circulating blood cells during PDAC progression in our KPC model and found that CD206-positive characteristics were acquired by macrophages in primary tumours when metastasis was detected and that BYL719 treatment prevented this behaviour. To better link these findings with the pro-metastatic effect of tumoural PI3K α , we identified the tumour cell / macrophage dialog that leads to promote tumour cell migration via increased TNF α secretion by macrophages initiated by tumoural PI3K α (New Fig. 7p-r). TNF α -induced migration of pancreatic cancer cells is prevented by BYL-719 treatment (New Fig. 7s); FGFR-inhibitor AZD4547 effect on migration was not additive (not shown). Hence, PI3K α pro-metastatic action acts both via a direct actin cytoskeleton remodeling and FGFR-induced tumour cell migration as well as via tumour-extrinsic promotion of TNF α secretion by macrophages and subsequent further activation of migratory phenotype. Interestingly, these data are in line with our early work (Guillermet J, PNAS 2003 and Bousquet C#, Guillermet-Guibert J# et al EMBO J 2006) that showed that class IA PI3K activity in pancreatic cancer cells is critical to activate NF- κ B activity preventing TNF α -induced cell death and promoting cell survival and migration. Finally, Fig. 1a showed that Hallmark TNF α signaling via NF κ B is one of the 6 hallmarks significantly changed in metastatic PDAC patients, further validating our finding.

Referee #2 (Comments on Novelty/Model System for Author):

Mouse model only very partially recapitulate the complexity of human PDACs

One of the advantages of KPC mice is that it reproduces the evolution of PDAC and gives access to less advanced stages that allow to dissect every step of the metastatic process (Embo Mol Med 2017; PMID: 28028012). These steps are difficult to study in this cancer usually diagnosed too late. For example, KPC model is used by others to describe novel modes of metastasis in PDAC (Embo Mol Med 2019, PMID: 30120146). In our study, we described for the first time the levels of circulating DNA in each step of pancreatic cancer development.

Referee #2 (Remarks for Author):

In this study Thibault et al propose a role for PI3K alpha in the control of PDAC cell migration, tumor progression and polarization of tumor-associated macrophages.

The rationale for the study is clear and some conclusions coherent with the experimental findings. However, the study presents several important weaknesses.

2.1. The differential enrichment of one specific PI3K-related pathway (Fig 1a,b) in metastatic vs. localized PDAC samples likely reflects the presence in this specific signature of genes that are not present in the other two related signatures in Fig. 1b. This should be investigated and explained.

We thank the reviewer for pointing this out. The difference refers to FGF7 and FGFR2 expression. FGFR signal activation was prevented by treatment with 2 μ M of AZD4547. AZD4547 significantly decreased R211 cell migration (New Fig. 2f). The concomitant inhibition of FGFR and PI3K α by simultaneous BYL-719 and AZD4547 treatment does not inhibit more cell migration than individual treatment which suggests that at least a part of PI3K α promigratory signal is due to FGFR activation. Interestingly, there is a large literature that shows that FGFR2 is promoting PDAC cell migration in vitro, in vivo and in PDAC patients (PMID: 18594526).

2.2. The links between the PI3K pathway and cell migration is far from being novel as there is a very extensive and mature literature in this area. In addition, since the PI3K pathway is downstream of activated KRAS and thus has an obvious impact on cell viability and proliferation, it is essential that the effects of these inhibitors on viability are carefully measured with assays more sensitive and specific than the MTT assay used here (e.g. in Fig 2f), such as Annexin V staining or equivalent. Finally, in migration assays it would be important to understand what are the properties of the cell lines analyzed. PANC1 have quasi-mesenchymal features and they are very different from other human PDAC cell lines. However, the features of the mouse cell lines used here and to what extent they have mesenchymal vs epithelial properties are unclear.

We agree that general links between PI3K pathway and migratory phenotypes are well known. PI3K isoform selectivity in cancer cell migration is not fully explored, and this topic is of critical importance. Pan-PI3K inhibitors failed in the clinics partly due to their increased toxicity. So far, the only PI3K inhibitors with medical authorization are isoform-selective.

To our knowledge, the demonstration that selective tumour-intrinsic PI3K α and not the other isoforms of PI3K uniquely drives pro-metastatic features in vitro and that it could be efficiently pharmacologically targeted in sophisticated preclinical models that present aggressive features is not available.

Mechanisms of isoform selectivity are also uncovered. Our data identify the production of a selective lipid metabolite PI-3,4,5-P₃ C36:2 produced by PI3K α (Figure 3); isoform selective production of specific PI-3,4,5-P₃ species is a novel mechanism of PI3K isoform selectivity (see also discussion).

Finally, we find that the efficiency of PI3K α targeting is not impacted by the heterogeneous genetic landscape of pancreatic cancer patients that activate PI3K pathway, which goes against the dogma described by Thorpe LM, Yuzugullu H et Zhao JJ, in Nature Review Cancer 2015 (PMID: 25533673). The model consists in predicting PI3K isoform specificity with the driving genetic alterations (i.e. the oncogenic Kras context show increased dependence to PI3K α , the loss of PTEN context increased dependence to PI3K β , the oncogenic PI3K α context increased dependence to PI3K α). We find that even pten-mutant PDAC cell line migration depends on

PI3K α activity. We sustain this result by demonstrating that basal level of PI3K α activation is sufficient to drive these pro-metastatic features (Figure 4); this result fits with the rare detection of oncogenic mutations in PI3K α encoding gene despite the crucial role of PI3K α activity in this cancer initiation (Baer et al, G&D 2014, PMID:25452273; Wu et al, Gastroenterology 2014, PMID: 25311989).

We assessed apoptotic (IncuCyte Annexin V) in R211 cells after 2-day treatment and found an expected increase of apoptotic cell death induced by PI3K α or pan-PI3K inhibitors (New Fig. 2h).

Epithelial or mesenchymal features of the murine cell line panel was assessed, and cells classified as either epithelial or mesenchymal phenotype (New Suppl Fig 9 g); importance of PI3K α activity for cell migration or survival was similar in both group (New Suppl Fig. 9h and i).

2.3. The interpretation of the links between PI3K, metastatization and detection of circulating tumor DNA is rather questionable. The amount of circulating tumor DNA correlates with the overall size of the tumor (including the primary tumor and metastases, if present), not necessarily with the metastatic behavior. In Fig. 5 the authors are comparing pT1N0 tumors with pT2N1 tumors, namely tumors with an overall low mass vs. tumors with an overall high mass. The higher level of ctDNA in the latter group is simply a reflection of this higher tumor mass, not of the different metastatic properties of these two tumors.

Our data are in line with the reviewer's comment: "A correlation was established between the proliferative index of cancer cells (assessed by ki67 index in primary tumour and metastatic sites) and levels of cfDNA, which reflects the global tumour burden (suppl. Fig. 13a)."

To answer this critics, we now compare T3 scored patients (N=6) with and without pathological nodal involvement (pT3N0, 2 patients versus pT3N1, 4 patients) (New fig. 5a,b). The whole paragraph was changed.

When we compared T3 scored patients with and without pathological nodal involvement (pT3N0, 2 patients versus pT3N1, 4 patients), we observed a stronger pAkt-Substrate IHC staining indicative of PI3K/Akt activity in the primary site in patients with tumour cell dissemination in lymph nodes. Among pT3N1 patients, those with a lower cfDNA had less pAKT substrate levels than those with higher cfDNA (New Fig. 5a,b).

We are convinced that, together with all the results gathered in the manuscript, these data provide sufficient ground to support the emergence of an extended translational study in PDAC patients; this is however out of scope of the current work and will be performed in a second step of our project in a prospective manner. It will also include analysis of inflammatory macrophage infiltration and of circulating cytokines levels in plasma. In resected PDAC (without residual macroscopic disease), ctDNA after surgery was demonstrated as a strong prognostic factor and was highly correlated with metastatic relapse, independently of the initial tumor burden. Those patients could benefit from any treatment that prevent evolution in macro-metastatic disease.

2.4. Links between PI3K and immune system. The difference in blood cell counts are of difficult interpretation in the context of metastatic KPC mice as they may represent indirect consequences of an advanced disease (with tumor necrosis etc.). More importantly, how the differences in IL3 expression would be linked to differences in stromal macrophages is completely unclear.

We agree with the reviewer, we verified in the first part of the figure (now Fig. 7) that inflammatory response occurs in our model; next, we find that BYL-719 treatment selectively decrease the number of CD206 positive macrophage, that are considered as inflammatory tumour-associated

macrophages TAM. We propose that this is a direct consequence of PI3K α inactivation in tumour cells. Indeed, conditioned medium from pancreatic tumour cells treated or not with BYL-719 or with genetic PI3K α inactivation were used to assess cytokine production of IC21 macrophage cell line (New Fig. 7p). Amongst tested cytokines, only TNF α production was significantly decreased by both pharmacological and genetic inactivation of PI3K α (New Fig. 7q,r). TNF α production is a marker of inflammatory tumour-associated macrophages TAM. IL-3 blocking Ab was added or not in the conditioned medium and reduced the level of secreted TNF α by IC21 cells (New Fig. 7r). We now show that TNF α promoted R211 cell migration and PI3K α inhibition by BYL-719 inhibited cell migration induced by TNF- α (New Fig. 7s). Hence, the tumour-macrophage dialog is possibly contributing to the anti-metastatic action of PI3K α inhibitor.

Our model is that PI3K α pro-metastatic action acts both via a direct actin cytoskeleton remodeling and FGFR-induced tumour cell migration as well as via tumour-extrinsic promotion of TNF α secretion by macrophages and subsequent further activation of migratory phenotype. Interestingly, these data are in line with our early work (Guillermet J, PNAS 2003 and Bousquet C#, Guillermet-Guibert J# et al EMBO J 2006) that showed that class IA PI3K activity in pancreatic cancer cells is critical to activate NF- κ B activity preventing TNF α -induced cell death and promoting cell survival and migration. Finally, Fig. 1a showed that Hallmark TNF α signaling via NFKB is one of the 6 hallmarks significantly changed in metastatic PDAC patients, further validating our finding.

30th Mar 2021

Dear Dr. Guillermet-Guibert,

Thank you for the submission of your revised manuscript to EMBO Molecular Medicine. We have now received the enclosed reports from the two referees who re-reviewed your manuscript. As you will see, they are supportive of publication, and I am therefore pleased to inform you that we will be able to accept your manuscript, once the following editorial points will be addressed:

1) Main manuscript text:

- Please answer/correct the changes suggested by our data editors in the main manuscript file (in track changes mode). This file will be sent to you in the next couple of days. Please use this file for any further modification.
 - Please remove the highlights in the text.
 - Please remove "Data not shown": As per our guidelines, all data referred to in the paper should be displayed in the main or Expanded View figures.
 - In the Discussion, please remove the reference to the Graphical Abstract.
 - Please replace "Methods" by "Material and Methods". Please include here most of the methods currently described in the Supplementary material and methods. Additionally:
 - o Human samples: please include a statement that informed consent was obtained from all subjects and that the experiments conformed to the principles set out in the WMA Declaration of Helsinki and the Department of Health and Human Services Belmont Report.
 - o Mice: please indicate the gender of the mice used in the experiments, please also indicate the housing and husbandry conditions.
- The tables in the Materials and Methods section should be moved to the end of the word file and referenced as Table files.
- Statistics: Please also indicate in the figures or in the legends the exact $n=$ and exact $p=$ values along with the statistical test used. You may provide these values as a supplemental table in the Appendix file.
 - Please add a "Data availability section": the primary datasets produced in this study need to be deposited in an appropriate public database and the accession numbers added to this section. Please also note that the datasets have to be made public before acceptance of the manuscript. If no new dataset was produced, please indicate: "This study includes no data deposited in external repositories"
 - Author contributions: the contribution of every author must be detailed in a separate section, after the acknowledgments.
 - Thank you for providing a "Conflict of Interest" section. Please move it after the author contribution.
 - Please update the reference format. References should be listed in alphabetical order and 10 authors should be listed before et al.
 - Please replace "Figure captions" by "Figure legends".
 - Please remove the section "Patient and public involvement": The funding by Fondation de France could be added to the acknowledgments, and the YouTube link could be provided in a "For more information" section, which is a dedicated space at the end of each article to list relevant web links for further consultation by our readers.
 - All supplementary information should be removed from the word Article file.

2) Figures, tables and appendix:

- Please note that we do not accept powerpoint figure files anymore. Please change the format accordingly.
- Please include a scale bar in Figure 2e. Please include a scale for Figure 6l.
- Please carefully check that all figures are referenced to in the main manuscript text (References to Fig.5 A-D and Fig. 6K-L are missing). Additionally, there are references in the main text to Figure 4, panels D-F, which do not exist.
- The "Supplementary Information" file should be made an "Appendix" file, which should start with a table of content. Appendix figures should be referred to in the main text as: "Appendix Figure S1, Appendix Figure S2" etc. As mentioned above, most of the methods should be described in the main manuscript file.

Furthermore, you have the possibility to make some figures "Expanded View (EV) Figures and tables that are collapsible/expandable online. A maximum of 5 EV Figures can be typeset. EV Figures should be cited as 'Figure EV1, Figure EV2' etc... in the text and their respective legends should be included in the main text after the legends of regular figures.

Additional Tables/Datasets should be labeled and referred to as Table EV1, Dataset EV1, etc. Legends have to be provided in a separate tab in case of .xls files. Alternatively, the legend can be supplied as a separate text file (README) and zipped together with the Table/Dataset file.

- Dataset EV legends: please include the file name and legend in the files.

3) Checklist:

Please provide a statement in section B/1/b, even if no statistical methods were used. Please also provide a statement in the section F/18 and 19. If no new dataset was generated, please indicate: This study includes no data deposited in external repositories

4) Thank you for providing source data. Please upload them as 1 pdf file per figure, with annotations where necessary.

5) Thank you for providing "The Paper Explained". I included minor edits (see below), please amend as you see fit. Please also modify the last paragraph to reflect the impact of your results rather than future direction.

PROBLEM: Pancreatic cancer is one of the most lethal solid cancers and is characterised by rapid progression after primary tumour detection. The key signalling events driving this fast evolution into macro-metastatic disease are still unknown.

RESULTS: Two unbiased approaches led to the identification of a high PI3K α activation signature in pancreatic primary tumours with bad prognosis. Our in vitro data showed that PI3K α is a major positive regulator of cancer cell escape from the primary tumour through actin cytoskeleton remodelling. PI3K α was inhibited in two preclinical models of pancreatic cancer. First, in a model of micro-metastatic disease (extra-pancreatic dissemination that goes undetected by ultrasound (US) imaging), mice presenting US-detected primary pancreatic tumours and increased circulating cell-free DNA (cfDNA) were treated. In a second model, tumour cell implantation and early proliferation in metastatic organs after intravascular injection were analyzed. A clinically relevant PI3K α - selective inhibitor (BYL-719/Alpelisib), currently tested in pancreatic cancer patients without patient stratification, was used in both models. Inhibition of PI3K α delayed primary tumour and micro-metastasis evolution, showing that PI3K α activity drove the evolution of micro-metastatic disease towards the macro-metastatic stage in these models. Mechanistically, tumour-intrinsic PI3K α activity increased pro- tumoural characteristics in peri-tumoural immune cells via increased IL-3

cytokine production. In return, inflammatory macrophages increased TNF α production, facilitating tumour cell migration.

IMPACT: Circulating tumour DNA represents a strong independent biomarker linked to relapse and poor survival in solid cancer patients. A clinical study in resected PDAC patients with micro-metastatic disease, characterised by high circulating tumoural DNA levels, is needed to assess if PI3K α -selective inhibitors could significantly prevent metastatic progression and death.

6) Thank you for providing a nice synopsis image. Please also provide a synopsis text. It should include a short stand first (maximum of 300 characters, including space) as well as 2-5 one-sentences bullet points that summarizes the paper. Please write the bullet points to summarize the key NEW findings. They should be designed to be complementary to the abstract - i.e. not repeat the same text. We encourage inclusion of key acronyms and quantitative information (maximum of 30 words / bullet point). Please use the passive voice.

7) As part of the EMBO Publications transparent editorial process initiative (see our Editorial at <http://embomolmed.embopress.org/content/2/9/329>), EMBO Molecular Medicine will publish online a Review Process File (RPF) to accompany accepted manuscripts.

This file will be published in conjunction with your paper and will include the anonymous referee reports, your point-by-point response and all pertinent correspondence relating to the manuscript. Please let us know whether you agree with the publication of the RPF and as here, **IF YOU WANT TO REMOVE OF NOT ANY FIGURES** from it prior to publication.

I look forward to receiving your revised manuscript.

Yours sincerely,

Lise Roth

Lise Roth, PhD

Editor

EMBO Molecular Medicine

To submit your manuscript , please follow this link:

Link Not Available

Photos 400-800 DPI

*Additional important information regarding figures and illustrations can be found at <https://bit.ly/EMBOPressFigurePreparationGuideline>

The system will prompt you to fill in your funding and payment information. This will allow Wiley to send you a quote for the article processing charge (APC) in case of acceptance. This quote takes into account any reduction or fee waivers that you may be eligible for. Authors do not need to pay any fees before their manuscript is accepted and transferred to our publisher.

***** Reviewer's comments *****

Referee #1 (Remarks for Author):

The authors have been responsive to the reviewers' critiques. I do not have any further comments.

Referee #2 (Remarks for Author):

In this revision the authors addressed some of the critical issues of the first version of the manuscript, notably the link between Akt activation and circulating tumor DNA as well as data on tumor infiltration by macrophages. I don't have any additional criticism or suggestion

The authors performed the requested editorial changes.

23rd Apr 2021

Dear Dr. Guillermet-Guibert,

Thank you very much for providing the files with the last requested changes. I am now pleased to inform you that your manuscript is accepted for publication and is now being sent to our publisher to be included in the next available issue of EMBO Molecular Medicine.

Congratulations on your interesting work,

With kind regards,

Lise Roth

Lise Roth, Ph.D
Editor
EMBO Molecular Medicine

Corresponding Author Name: Julie Guillermet-Guibert
Journal Submitted to: EMBO Mol Med
Manuscript Number: EMM-2020-13502